# Prostate cancer reshapes the secreted and extracellular vesicle urinary proteomes

Amanda Khoo[1,2,17], Meinusha Govindarajan[1,2,17], Zhuyu Qiu [3,4,5,6,17], Lydia Y. Liu [1,2,3,4,5,6], Vladimir Ignatchenko[2], Matthew Waas[2], Andrew Macklin[2], Alexander Keszei[2], Sarah Neu[7], Brian P. Main[8], Lifang Yang[8,9], Raymond S. Lance[10], Michelle R. Downes[11,12], O. John Semmes [8,9], Danny Vesprini[13,14], Stanley K. Liu [1,13,14], Julius O. Nyalwidhe[8,9], Paul C. Boutros [1,3,4,5,6,15,16] ✉ & Thomas Kislinger [1,2] ✉

Urine is a complex biofluid that reflects both overall physiologic state and the state of the genitourinary tissues through which it passes. It contains both secreted proteins and proteins encapsulated in tissue-derived extracellular vesicles (EVs). To understand the population variability and clinical utility of urine, we quantified the secreted and EV proteomes from 190 men, including a subset with prostate cancer. We demonstrate that a simple protocol enriches prostatic proteins in urine. Secreted and EV proteins arise from different subcellular compartments. Urinary EVs are faithful surrogates of tissue proteomes, but secreted proteins in urine or cell line EVs are not. The urinary proteome is longitudinally stable over several years. It can accurately and non-invasively distinguish malignant from benign prostatic lesions and can risk-stratify prostate tumors. This resource quantifies the complexity of the urinary proteome and reveals the synergistic value of secreted and EV proteomes for translational and biomarker studies.

Human urine is produced when blood is filtered by the kidneys. This filtration retains proteins and nutrients, removes undesired metabolites, and regulates pH and water levels. Excreted urine is less than 1% of pre-filtered volume[1]. After filtration, urine is stored in the bladder for minutes to hours before being voided through the urethra. In males, the urethra runs through the prostate, a gland that produces prostatic

fluid[2]. Because urine can spend significant residence time within the genitourinary tract, it accumulates bioanalytes that reflect the current state of those tissues[3]. The urine is therefore a remarkably complex biofluid and provides a non-invasive snapshot of both organismal state and genitourinary tissues. Its molecular composition can vary across individuals, and within a single individual over time[4]. Urine has been

[1]Department of Medical Biophysics, University of Toronto, Toronto, ON M5G 1L7, Canada. [2]Princess Margaret Cancer Centre, University Health Network, Toronto, ON M5G 2C1, Canada. [3]Jonsson Comprehensive Cancer Center, David Geffen School of Medicine, University of California, Los Angeles, Los Angeles, CA 90024, USA. [4]Department of Human Genetics, University of California, Los Angeles, Los Angeles, CA 90095, USA. [5]Department of Urology, University of California, Los Angeles, Los Angeles, CA 90095, USA. [6]Institute for Precision Health, David Geffen School of Medicine, University of California, Los Angeles, Los Angeles, CA 90095, USA. [7]Division of Surgery, Urology, Sunnybrook Health Sciences Centre, Toronto, ON M4N 3M5, Canada. [8]Leroy T. Canoles Jr. Cancer Research Center, Eastern Virginia Medical School, Norfolk, VA 23507, USA. [9]Department of Microbiology and Molecular Cell Biology, Eastern Virginia Medical School, Norfolk, VA 23507, USA. [10]Spokane Urology, Spokane, WA 99202, USA. [11]Division of Anatomic Pathology, Laboratory Medicine and Molecular Diagnostics, Sunnybrook Health Sciences Centre, Toronto, ON M4N 3M5, Canada. [12]Laboratory Medicine and Pathobiology, University of Toronto, Toronto, ON M5S 1A8, Canada. [13]Department of Radiation Oncology, University of Toronto, Toronto, ON M5T 1P5, Canada. [14]Odette Cancer Research Program, Sunnybrook Research Institute, Toronto, ON M4N 3M5, Canada. [15]Department of Pharmacology and Toxicology, University of Toronto, Toronto, ON M5S 1A8, Canada. [16]Broad Stem Cell Research Center, University of California, Los Angeles, CA 90095, USA. [17]These authors contributed equally: Amanda Khoo, Meinusha Govindarajan, Zhuyu Qiu. ✉e-mail: PBoutros@mednet.ucla.edu; thomas.kislinger@utoronto.ca

widely proposed as a non-invasive longitudinal biomarker matrix[5,6], and specific DNA, RNA, or protein species have been identified in urine that can serve as biomarkers[7–10].

Proteins enter the urine in two ways: leakage at the glomeruli of the kidney and throughout the urogenital tract. It is believed that the vast majority of the excreted urinary proteome derives from tissues of the genitourinary tract, rather than leakage from the kidneys. Urogenital proteins can enter the urine via passive release through cell death, through active translocation, and as part of secreted extracellular vesicles (EVs)[11,12]. EVs are nanosized particles with a lipid bilayer released by cells into the extracellular milieu. They vary dramatically in size, ranging from 30–2000 nm in diameter, and are heterogeneous in their mechanisms of biogenesis, molecular composition, and function[13]. EVs play a crucial role in both physiology and in the pathogenesis of diseases, including cancer[14]. EV and secreted proteomes are hypothesized to be context-driven and tissue-specific[15], but their presence, population variability, and disease relevance in urine remain poorly characterized.

To fill this gap, we generated comprehensive urinary proteomic profiles from 190 treatment-naïve men with a range of benign and malignant conditions. In this work, we demonstrate a simple protocol that uses urine to directly sample prostate proteins. This allows us to identify the tissue and subcellular origins of urinary proteins and EVs, and to quantify how the urine proteome changes over time in specific individuals. Urinary EVs, but not those released from prostate cancer cell lines nor secreted urinary proteins, accurately reflect prostatic tissue. Prostate tumor-specific urinary proteins accurately distinguish men with and without prostate cancer and risk-stratify those already with the disease. Canonical EV markers are not effective in urine, but we identify context-dependent urine EV cargo that accurately marks specific urinary EV populations.

## Results

### Digital rectal examination enriches for prostate proteins

The urine proteome is believed to derive almost exclusively from tissues of the genitourinary tract: the kidney, bladder, and (in men) prostate[11]. Perturbation of the prostate gland using digital rectal examination (DRE) enriches for prostate-specific RNAs in urine[16]. While the mechanism is unknown, it is thought to occur by expelling prostatic fluid into the urethra, where it can be collected as part of first-catch urine. DREs are routine, minimally-invasive physical manipulations performed millions of times annually by oncologists and primary care physicians and thus might provide a simple approach to enrich prostate-derived proteins in urine.

We therefore collected matched pre- and post-DRE urines from ten men (Fig. 1a and Supplementary Data 1) and applied differential ultracentrifugation to separate urine-soluble proteins (uSP) from urinary extracellular vesicles (uEVs). We further isolated two uEV populations based on size[17]: one at $20,000 \times g$ (termed uEV-P20) and the other at $150,000 \times g$ (uEV-P150; Supplementary Fig. 1a). To determine if a DRE influenced the biophysical characteristics of EVs, we quantified EV diameter, number, and morphology by nanoparticle tracking and by transmission electron microscopy (Fig. 1b and Supplementary Fig. 1b–d). uEV biophysical characteristics were unchanged before or after a DRE (Supplementary Fig. 1b–d).

To evaluate if a DRE increased the abundance of prostate-derived proteins, we measured the proteomes of each urine fraction using mass spectrometry (Supplementary Data 2). Prostate tissue-derived proteins were curated from three independent tissue proteomics datasets[18–20], then annotated in urine. While the total number of urine proteins was unchanged by a DRE ($4064 \pm 604$ pre-DRE vs. $4362 \pm 511$ post-DRE; $P = 0.31$; two-sided Wilcoxon signed-rank test), prostate tissue-derived proteins were significantly more abundant in post-DRE urine ($P < 2.2 \times 10^{-16}$; Fig. 1c–e). These included classic prostate marker proteins like PSMA (*FOLH1*) and PSA (*KLK3*)[20] (Fig. 1f and

Supplementary Fig. 1e). Thus, a DRE significantly enriches the urine for proteins of prostate origin but does not influence EV biophysical characteristics, suggesting the latter may be relatively tissue-independent.

### The subcellular origins of urinary proteins

To investigate urinary protein heterogeneity, we next collected post-DRE urines from 190 men: 64 men with no cancer diagnosis and 126 with untreated prostate cancers. The prostate cancer patients reflected the full risk spectrum of primary disease, with biopsy ISUP Grade Groups (GG) ranging from low (GG 1) to high (GG 5; Fig. 2a). We isolated uSP, uEV-P20 and uEV-P150 from post-DRE urine and quantified their proteomes (Fig. 2b; Supplementary Data 1). uEV biophysics and urine protein counts were largely independent of disease status, age or serum PSA levels (Supplementary Fig. 2a–g). We observed an increasing trend in uEV particle count in cISUP GG 1–3 followed by a decrease in particle count in cISUP GG 4–5, although this finding requires further validation in larger cohorts due to the low numbers of patients in each group (Supplementary Fig. 2c). The uEV-P20 and uEV-P150 fractions were biophysically similar in size morphology and particle count (Fig. 2c–e).

Despite these biophysical similarities, the proteomes of different urine fractions were significantly different. The uEV-P20 fraction had the most detectable proteins (Fig. 2f) and was the most biophysically and proteomically diverse, suggesting heterogeneity in vesicular type or origin (Fig. 2g). uEV isolation was highly reproducible, with larger proteomic differences observed between patients and clinical groups compared to within samples (Supplementary Fig. 3a, b). Of the 6518 proteins detected in uEVs, 60% were identified as EV cargo in previous studies (Supplementary Fig. 3c, d)[17,21–24]. uEV fractions were more similar to one another than to the soluble protein fraction, consistent with the detected EV proteins being true cargo rather than co-isolated urinary proteins (Supplementary Fig. 3e–g). To determine the differential origins and biology represented by each fraction, we performed differential proteome analysis. We identified proteins detected in one fraction but not the others (Fig. 2h) and proteins detected at different abundances across fractions (Fig. 2i). Each fraction was defined by presence of ~50 proteins and differential abundance of ~500 others (Fig. 2j). Fraction-specific proteins tended to arise from specific subcellular compartments (Supplementary Data 3). Urine-soluble proteins were typically secreted or derived from the Golgi apparatus, while uEV-P20 proteins derived from mitochondria or endoplasmic reticulum and uEV-P150 proteins from the plasma membrane (Fig. 2k).

### uEVs but not cEVs reflect the prostate tissue proteome

We next quantified how well each of the three urine fraction proteomes reflects the proteome of prostate tissue[18,19]. A majority (67%) of all proteins detected in prostate tissue were detected in one or more of the three urine fractions (Fig. 3a). EV fractions were a much richer source of prostate-derived proteins than urine-soluble proteins. Only 116 prostate-derived proteins were identified in unfractionated urine, whereas 2439 were identified only in one or both of EV fractions and 4968 in both EV and non-EV urine. Protein abundances were well-correlated between urinary and tissue proteomes (Fig. 3b), with uEV-P20 being the best surrogate for prostate tissue.

To further quantify the tissue provenance of urinary proteins, we used RNA-seq data from normal tissues (GTEx[25]) and from normal tissue adjacent to tumors (NAT, TCGA[26–28]) to identify transcripts enriched in prostate, kidney or bladder (Fig. 3c). Differentially abundant transcripts were then used as signatures of tissue origin to quantify the contribution of each tissue to each sample. In uEVs, prostate proteins were very significantly more abundant than non-prostate proteins ($P < 1 \times 10^{-5}$). In contrast, the soluble urine fraction (uSP) showed the inverse trend: it was significantly depleted in prostate-derived proteins ($P < 2.2 \times 10^{-16}$; Fig. 3d). The soluble protein

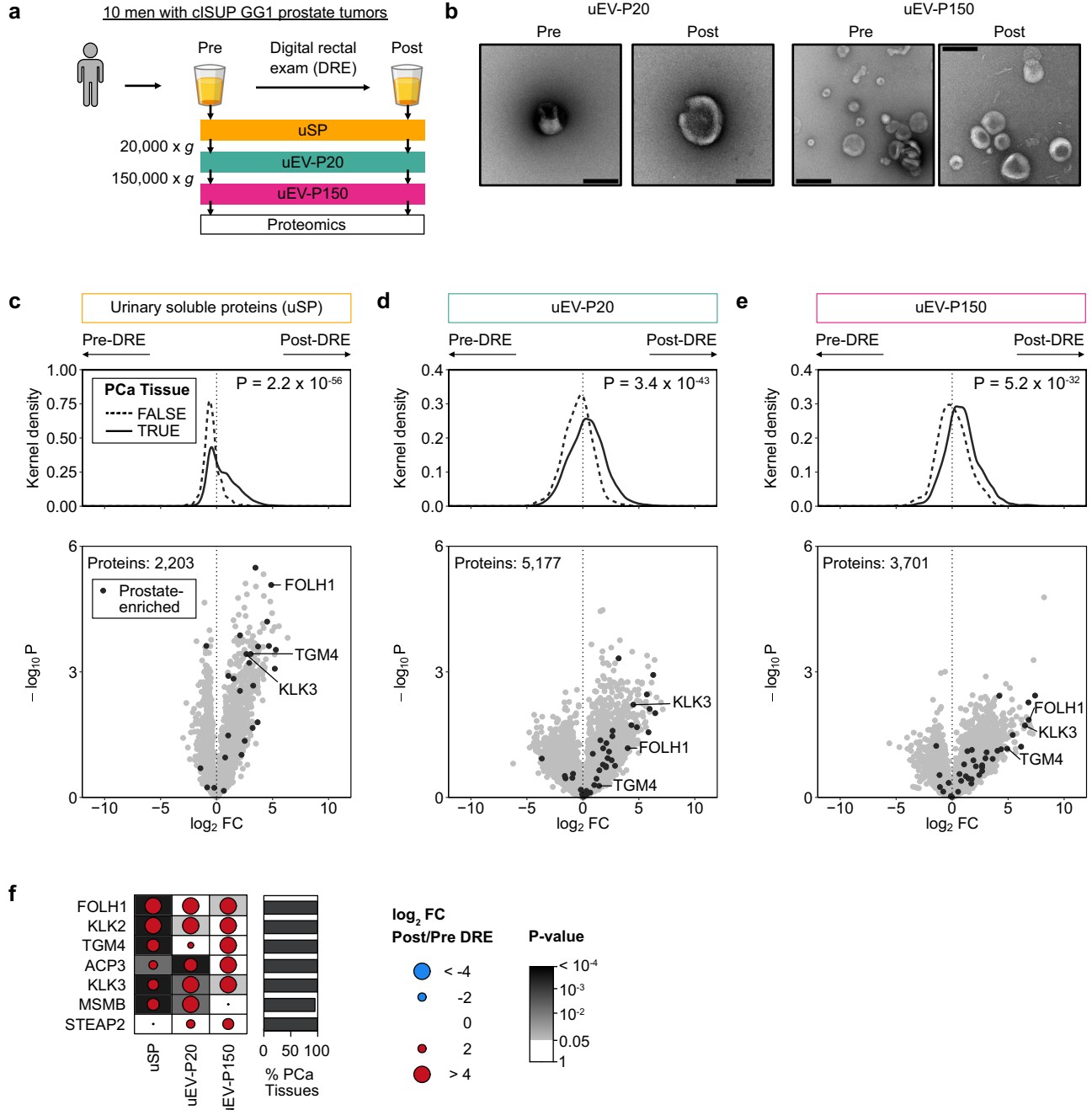

**Fig. 1 | Digital rectal exam enriches for prostate tissue-derived proteins.**
**a** Matched pre- and post-DRE urine proteomes from clinical International Society of Urological Pathology (cISUP) Grade Group (GG) 1 patients consisting of unfractionated urine (soluble proteins, uSP) and two subtypes of urinary extracellular vesicles (uEV) isolated by differential ultracentrifugation at 20,000 × $g$ (uEV-P20) and 150,000 × $g$ (uEV-P150). **b** Transmission electron microscopy images of uEVs isolated from pre- and post-DRE urine from a single individual. Scale bar: 200 nm. Images are representative of three biological replicates. **c**–**e** Proteomic differences in pre- and post-DRE urines in uSP (**c**), uEV-P20 (**d**) and uEV-P150 (**e**) fractions. Top panel: log₂ fold change (log₂FC) in protein abundance in urine, grouped by protein detection in prostate tissues[18,19]. Bonferroni-corrected *P*-values from two-sided Mann−Whitney *U* tests. Bottom panel: log₂FC in protein abundances. Prostate-specific proteins as per the Human Protein Atlas are in black. **f** Differences in pre- vs. post-DRE urine (log₂FC) for select prostate tissue-specific proteins[20]. Percentage of 157 prostate cancer (PCa) tissues[18,19] in which each protein was detected on the left. Background shading denotes *P*-value < 0.05 from a two-sided Wilcoxon signed-rank test. Source data are provided as a Source Data file.

fraction (uSP) was enriched in functions classically associated with blood (lipoprotein, blood microparticle; Fig. 3e). The soluble proteome was also highly enriched in cell surface proteins, likely from the shedding of extracellular domains[29,30]. Proteins involved in multivesicular body biogenesis were over-represented in uEV-P150, suggesting an enrichment of exosomes[31]. Consistent with the univariate protein analysis, uEV-P20 fraction more closely reflected prostate tissue, harboring proteins originating from mitochondria, ribosome, and extracellular matrix[18]. These data suggest that the prostate primarily sheds EVs into urine and that these can provide a robust non-invasive proxy of its proteome.

Cell line conditioned media has been widely used to study secreted cellular components[32,33]. To ascertain if cell line-derived EVs (cEVs) accurately reflect tumor tissue, we isolated and proteomically

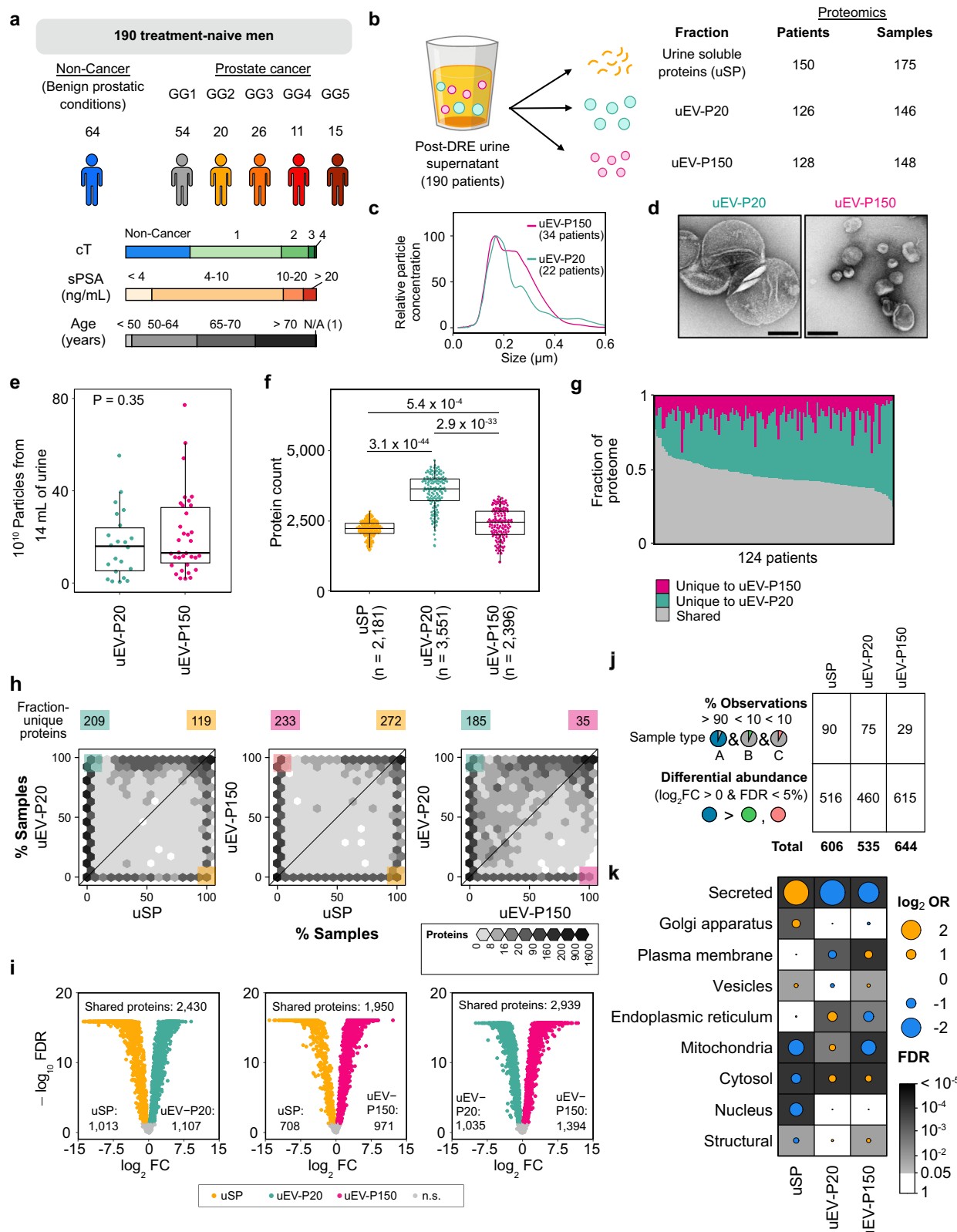

characterized cEVs from five prostate cell lines (Fig. 4a). EVs from urine and from cell line conditioned media displayed similar biophysical characteristics (Supplementary Fig. 4a–d). More proteins were detected in cEV-P20 than in cEV-P150, but fewer than in whole-cell lysates (Supplementary Fig. 4e, f). Cell line whole-cell lysates closely resembled tumor tissue in both protein composition (Fig. 4b) and abundance (Fig. 4c). By contrast uEVs and cEVs differed, particularly in abundance

(Fig. 4d, e). Urinary EVs were better surrogates for prostate tissue than EVs derived from cell lines (Supplementary Fig. 4g). Commonly used EV markers like CD9, CD81 and CD63[34,35] were amongst the proteins more abundant in EV-P150 than EV-P20 in both cell lines and urines (Fig. 4f and Supplementary Fig. 4h), while other EV markers like FLOT1 were discordant between cell lines and urine (Fig. 4f). PSA protein expression in whole-cell lysates was consistent with the androgen

**Fig. 2 | Post-DRE uEV fractions harbor distinct protein cargo. a** Cohort overview. Clinical ISUP Grade Group (GG); cT: clinical T category; sPSA: Serum Prostate-Specific Antigen (ng mL$^{-1}$); Age: Age at diagnosis (years). **b** Post-DRE urine fractions analyzed by proteomics, total patients, and samples per fraction. **c** Distribution of particle sizes determined by nanoparticle tracking analysis, mean of $n = 22$ (uEV-P20) and $n = 34$ (uEV-P150). **d** Representative transmission electron microscopy images of uEV fractions from one cISUP GG 1 patient. Scale bar: 200 nm. Images are representative of three biological replicates. **e** Particle concentration for uEV-P20 and uEV-P150 from 22 and 34 patients, respectively. **f** Number of proteins quantified by mass spectrometry. Dots represent samples (Samples: 175 uSP, 146 uEV-P20, 148 uEV-P150). *P*-values for (**e**, and **f**) from two-sided Mann–Whitney *U* tests. **g** Fraction of uEV-P20 and uEV-P150-unique proteins from 124 patients with matched uEV fractions. **h** Number of samples each protein was detected in ($n = 96$

patients). For each pairwise comparison, the numbers of proteins present in >90% of samples in one sample type and <10% of the other are labeled on top of each panel. **i** Differences in shared protein abundance between fractions. Significant differences (FDR < 0.05, two-sided Wilcoxon signed-rank test) are in green (uEV-P20), pink (uEV-P150) or yellow (uSP). n.s.: non-significant. Total differentially abundant proteins in bottom corners. $n = 96$ patients. **j** Fraction-enriched proteins either unique to one fraction or differentially abundant in one fraction relative to the other two. **k** Odds ratio (OR) of gene set enrichment for each subcellular localization[62] for proteins from (**j**). Grey background shading indicates FDR < 0.05 (Fisher's exact test). Boxplots are shown with the line indicating the sample median, the box indicating the 25th and 75th percentiles, and the whiskers indicating the ±1.5 × interquartile range (IQR). Source data are provided as a Source Data file.

sensitivity of the cell lines, with androgen insensitive cell lines DU145, PC3 and RWPE1 not expressing PSA and androgen-sensitive cell lines 22Rv1 and LNCaP expressing PSA (Supplementary Data 4)[36]. In cEVs, only EVs isolated from LNCaP had detectable PSA. Mitochondrial proteins were over-represented in cEV-P20 and plasma membrane proteins in cEV-P150 (Fig. 4g). Thus, post-DRE EV-associated proteins more accurately reflect prostate tissue than prostate cancer cell line EVs or, particularly, post-DRE urine-soluble proteins.

## Biomarker potential of urinary proteomes

For a biomarker to be useful, it needs to be robust to a variety of types of errors. While urine is not prone to spatial biases, it is unclear to what extent an individual's urine proteome is stable over time. To quantify the temporal stability of post-DRE urine, we evaluated longitudinal samples in five prostate cancer patients over several years. From each patient, we collected post-DRE urine at multiple time points; all patients were managed by active surveillance without any indication of clinical progression (Fig. 5a; Supplementary Data 1). We used variance analysis to quantify which proteins were more variable between samples of a single patient (intra) or across individuals (inter). On average, proteomes were more similar within patients than between them, for both EV cargo and urine-soluble proteins (Fig. 5b). To identify proteins that might be particularly useful as biomarkers, we identified those that were longitudinally stable using the intraclass correlation coefficient (ICC)[37]. The higher a protein's ICC, the less its variance in protein abundance is caused by random fluctuations over time. A subset of proteins was highly longitudinally stable in each fraction (Supplementary Fig. 5a), and these comprise excellent candidate biomarkers because they are robust to physiological variability over several years.

Next, we sought to evaluate the biomarker potential of urine-soluble and uEV proteins for predicting prostatic disease. Men with benign prostatic conditions (non-cancer [NC]) included individuals with elevated serum PSA levels and benign prostatic hyperplasia as well as patients with no diagnosed prostate cancer on transrectal ultrasound-guided 12-core biopsy (Supplementary Data 1). Patients with and without prostate cancer had similar serum PSA abundances ($P = 0.34$). Despite this similarity, thousands of soluble (Fig. 5c) and EV proteins (Fig. 5d, e) differed between patients with and without cancer. The specific differentially abundant proteins varied between urinary fractions (Supplementary Fig. 5b). Of these proteins that were differentially abundant in prostate cancer and non-cancer uEVs, 541 proteins were uniquely detected in our dataset compared to other datasets of post-DRE urine-derived uEVs from patients with prostatic disease[17,21] (Supplementary Data 4). In addition, only 21 proteins were unique to a single disease subgroup (i.e. non-cancers or specific cISUP GG). These results suggest that differences in disease groups are reflected in the differential abundance of proteins in the uEV proteome.

Pathways associated with malignancy in prostate tissue also differed from those in urine (Supplementary Fig. 5c). For example, proteins regulated by androgen response, such as PSA, had increased abundance in tumor tissue compared to NAT tissue, but had the

opposite trend in urine-soluble proteins and uEVs. Consistent with other studies[38,39], PSA (*KLK3*) protein abundance was consistently significantly reduced in all fractions of urine from prostate cancer patients relative to men without a cancer diagnosis, despite being increased in serum and in tumor regions[19] (Fig. 5c–e). Therefore, our data suggest that pathways associated with malignancy are reflected in different subsets of the prostate urinary proteome.

To create biomarkers of prostatic disease, we focused on proteins that were frequently detected (in >50% of samples), enriched in a urinary fraction, prostate-derived, and longitudinally stable (ICC > 0.4) (Fig. 6a). This filtering strategy retained 226 uSP proteins, 280 uEV-P20 proteins, and 235 uEV-P150 proteins. We used statistical machine learning to create and validate classifiers independently for each urine fraction. First, we created classifiers that distinguish cancer from non-cancer based solely on urine proteins; these had AUCs ranging from 0.92–1, significantly outperforming serum PSA (Fig. 6b). In an independent, prospective validation cohort of 30 patients with proteomic data from all three urinary fractions (Supplementary Data 1), proteins had concordant effect sizes and validation AUCs ranging from 0.71–0.81 (Fig. 6c, d). Next, we applied the same methodology to the more challenging task of distinguishing low-grade from high-grade cancer. This is an important clinical question, as low-grade disease is typically managed by active surveillance, and higher-grade disease by definitive local therapy. Urine distinguished low- from high-grade prostate cancer with AUCs ranging from 0.73–0.79 (Supplementary Fig. 6a), again matching or exceeding serum PSA. In two independent validation cohorts of 199 and 75 patients (Supplementary Data 1), effect sizes of uSP proteins between the discovery and validation cohorts were largely concordant (Supplementary Fig. 6b). Intriguingly, signatures of grade also informed on disease status, reflecting an overlap in determinants of initiation and progression, as seen in studies of prostate cancer genetic drivers[40] (Supplementary Fig. 6c). Proteins within these signatures exhibit longitudinal stability and were univariately associated with disease status (Supplementary Fig. 6d). These data suggest that the urine proteome is an untapped source of biomarkers for genitourinary disease.

## Markers of prostate tumor uEVs

Post-DRE urine can non-invasively sample the prostate tissue proteome and has significant biomarker potential. However, the soluble fraction and the protein cargo of different-sized EVs differ in their origin and association with disease phenotypes. As examples, SPOCK1 protein was significantly lower in the soluble fraction of urine from cancer patients but not in uEVs, while PCYOX1 shows the inverse (Supplementary Fig. 6d). This interplay between subcellular origin, clinical phenotype, and urine fraction is summarized in Fig. 7a and suggests that protein cargo is selectively packaged into EVs in processes that are dysregulated during malignant transformation (Supplementary Fig. 7a).

This heterogeneity in origins and association with clinico-epidemiologic characteristics highlights the importance of

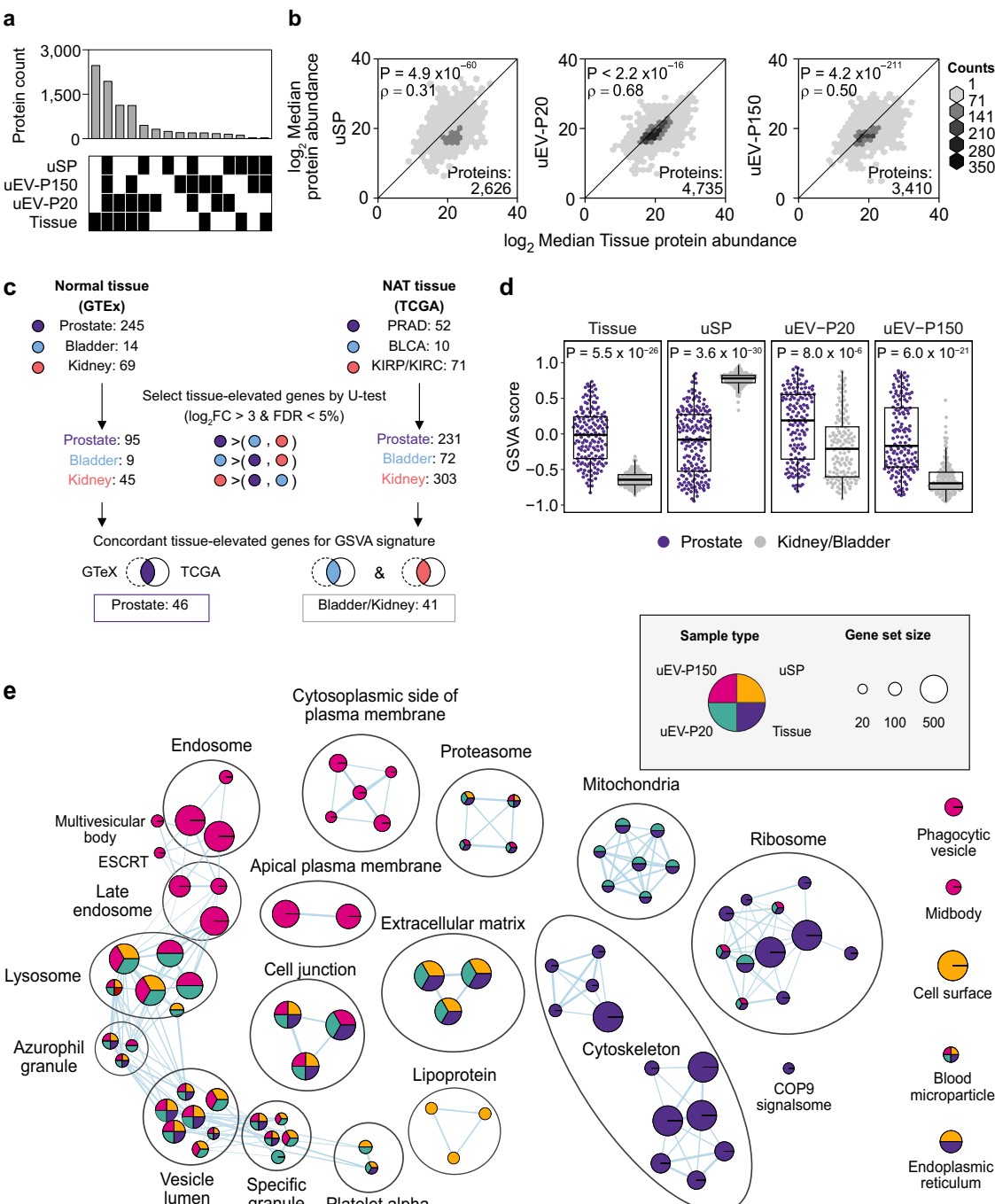

**Fig. 3 | uEV proteome closely reflects the prostate tissue proteome. a** Overlap in proteins quantified in each sample type. Samples: Urinary soluble proteins (uSP) = 175, uEV-P150 = 148, uEV-P20 = 146, Tissue = 157; Proteins: uSP = 3150, uEV-P150 = 3878, uEV-P20 = 5462, Tissue = 7438. **b** log$_2$ median protein abundance between prostate tissue and uSP (left), uEV-P20 (middle), and uEV-P150 (right). Spearman's rank correlation and its $P$-value (two-tailed) are shown. **c** Analysis strategy to identify tissue-associated genes in RNA-seq of normal or normal adjacent to tumor (NAT) prostate, bladder, and kidney from the Genotype-Tissue Expression project (GTEx)[25] or The Cancer Genome Atlas (TCGA)[26–28] (Supplementary Data 4). $P$-values from the two-sided Mann–Whitney $U$ test adjusted using the Benjamini–Hochberg method (FDR). **d** Gene set variation analysis scores (GSVA) of sample types based on tissue-specific signatures (prostate: purple, kidney/bladder: grey). Samples: uSP = 175, uEV-P150 = 148, uEV-P20 = 146, Tissue = 157. $P$-values from two-sided Mann–Whitney $U$ test. Boxplots are shown with the line indicating the sample median, the box indicating the 25th and 75th percentiles, and the whiskers indicating ±1.5 × IQR. **e** Gene Ontology: Cellular Component (GO:CC) gene sets over-represented in each sample type (Supplementary Data 3). Only significant gene sets (g:SCS-adjusted $P$-value < 0.05 from a Fisher's one-tailed test in g:Profiler[64]) are visualized. Source data are provided as a Source Data file.

accurately isolating specific EV populations. The gold-standard intensive centrifugation used here is expensive and time-consuming, so affinity-based isolation methods are strongly preferred for rapid translational and clinical studies. Affinity methods require the identification of proteins characteristic of specific EV subpopulations[41]. We

evaluated the performance of 13 EV markers that are ubiquitous to multiple human cancer tissues and fluids but not yet evaluated in prostate urinary EVs[41]. Four of these were elevated in uEV-P150 relative to soluble urine proteins and five were elevated in uEV-P20. Others were depleted in uEVs, and none showed the large effect sizes or small

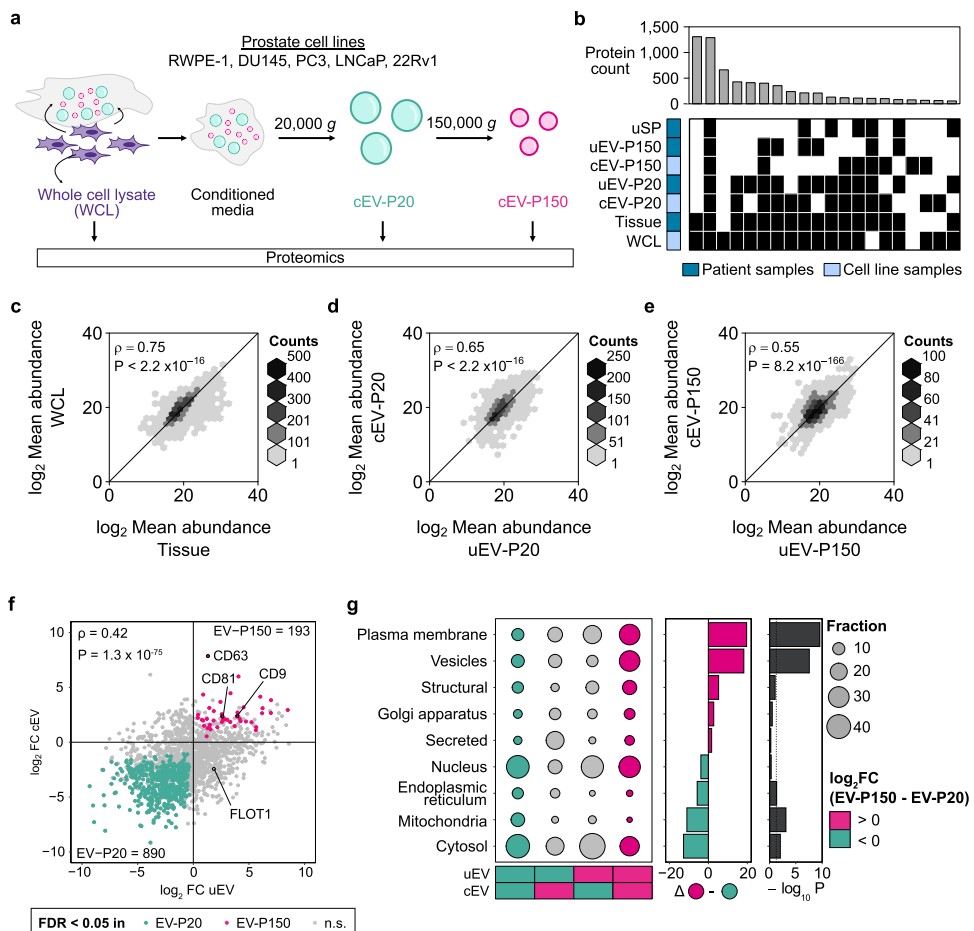

**Fig. 4 | Prostate cancer cell line EVs do not fully reflect prostate fluid EVs.**
**a** Overview of cell line EV (cEV) isolation from conditioned media. **b** Overlap in proteins quantified between sample types. **c**–**e** Spearman's correlation between $\log_2$ mean protein abundance in patient and cell line fractions. Cell line protein abundances are the mean of three experimental replicates from five cell lines.
**f** $\log_2$FC between EV-P150 and EV-P20 in uEV (x-axis) vs. cEV (y-axis). Significantly differentially abundant proteins in both uEV and cEV (two-sided Mann–Whitney $U$

test FDR < 0.05) are pink (more abundant in EV-P150) or green (more abundant in EV-P20). Grey dots represent non-significant proteins (n.s., FDR ≥ 0.05).
**g** Organellar enrichment for proteins (two-proportion $z$-test) in each quadrant from (**d**) (left panel), annotated with differences in fraction (middle panel) between EV-P150 (top right quadrant) and EV-P20 (bottom left quadrant) and $P$-value (right panel). Source data are provided as a Source Data file.

inter-sample variability needed for ideal affinity-based markers (Fig. 7b). Thus, canonical EV protein markers do not appear to be optimal for uEV identification and isolation.

We therefore sought to identify protein markers to distinguish prostate-derived uEVs. We selected proteins that were both distinct to one urine fraction (Fig. 2j) and were known to be present in prostate tissue. We separated these into three protein subsets: those associated with presence of a tumor, those associated with tumor grade and disease-invariant "core" proteins (Fig. 7c). These subsets were functionally distinct, with prostate-specific secreted proteins such as KLK3 and ACPP (Cluster $C_{uSP}8$) being specific to the urine-soluble protein fraction but not uEVs, and GTP binding proteins specific to the uEV-P150 fraction (Cluster $C_{P150}8$)[15] (Supplementary Fig. 7b–d).

Finally, to identify specific actionable markers for uEV affinity studies, we selected predicted cell surface proteins from each subset[42] (Fig. 7c and Supplementary Fig. 7e, f). The resulting five uEV-P20 and ten uEV-P150 tumor markers that differed in patients with cancer and benign disease include the classical EV marker CD63[34,35] (Fig. 7d). Grade markers (Supplementary Fig. 7e) contain markers such as ITGB2 and SLC4A1 in the uEV-P20 fraction. Disease-invariant core markers of uEV-P150 include classical EV markers CD9 and CD81, which were frequently detected in uEV-P150 (Supplementary Fig. 7f). These data indicate that the EV proteome is context-specific (Supplementary

Fig. 7g), and that protocols for rapid, specific isolation of EV sub-populations may differ from those useful in plasma or some other tissues.

## Discussion

Urine contains a complex mixture of proteins that differ in their form of release and tissue of origin, resulting in a dynamic range of concentrations spanning ten orders of magnitude[4]. To better define the urinary proteome, we used fractionation to distinguish urine-soluble proteins from proteins carried in urinary extracellular vesicles (uEVs). uEVs of different sizes and densities contained proteins from different subcellular origins, suggesting distinct biogenesis[43,44]. uEVs, particularly the uEV-P20 population, appear to derive heavily from prostate tissue, and to accurately reflect its proteome. By contrast, neither the soluble urine proteome nor cell line-derived EVs did so. This highlights the need to prioritize patient-derived EVs in translational studies. This is particularly key for diseases where models that faithfully recapitulate aspects of the natural history of cancer are lacking, such as hormone sensitivity and hypoxia in prostate cancer.

Prior work has rigorously quantified the role of factors such as sample collection, processing[45,46], and storage[47] on urine proteomes. We show that digital rectal examinations are a simple way to enrich for prostate proteins and EVs in first-catch urine. While post-DRE urine

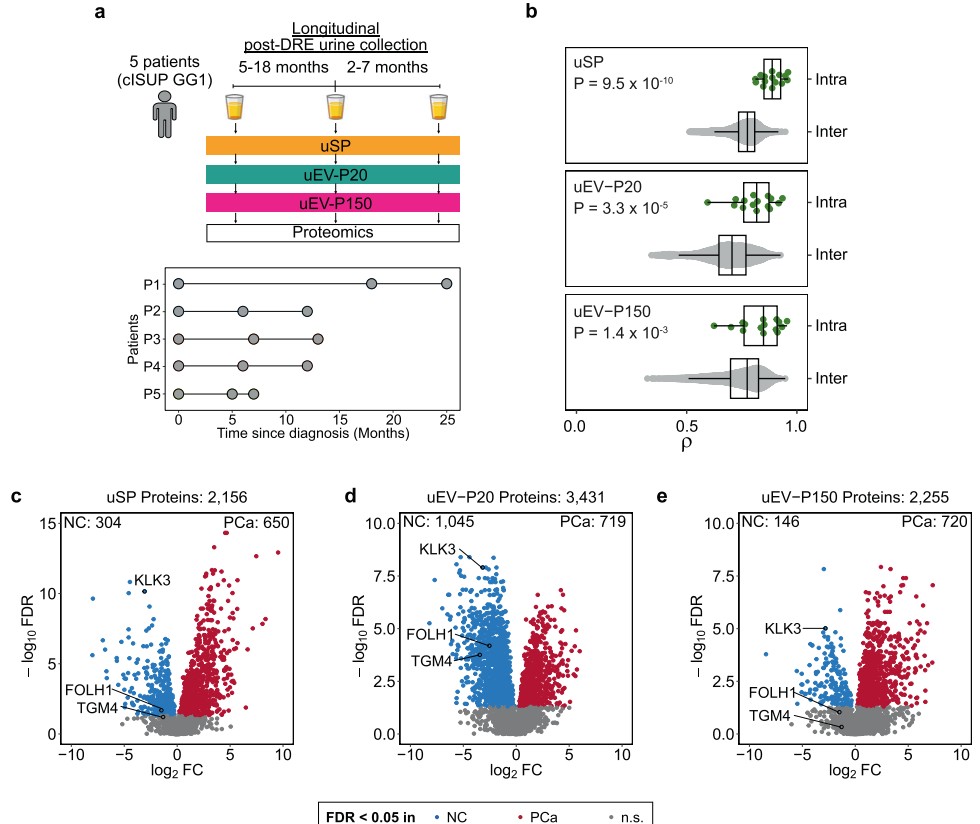

**Fig. 5 | uEV proteome is temporally stable and reflects clinical behavior. a** Time points for each post-DRE urine collection in a cohort of five patients with cISUP Grade Group (GG) 1 tumors on active surveillance that did not upgrade over the course of urine collection. **b** Correlation in protein abundance (Spearman's ρ) within individuals (Intra, green, 5 patients with cISUP GG 1 tumors), and between individuals (Inter, grey; patients: uSP = 150, uEV-P20 = 126, uEV-P150 = 128). *P*-values from two-sided Mann−Whitney *U* tests. Boxplots are shown with the line indicating the sample median, the box indicating the 25th and 75th percentiles, and the whiskers indicating ±1.5 × IQR. **c**−**e** Significantly differentially abundant proteins between prostate cancers (PCa, red) vs. non-cancers (NC, blue) using a two-sided Mann−Whitney *U* test FDR < 0.05. n.s.: Non-significant, grey. NC patients for uSP and uEV are not matched. Patients: $uSP_{NC} = 39$, $uSP_{PCa} = 136$, $uEV\text{-}P20_{NC} = 22$, $uEV\text{-}P20_{PCa} = 132$, $uEV\text{-}P150_{NC} = 25$, $uEV\text{-}P150_{PCa} = 131$. Source data are provided as a Source Data file.

proteomes can vary over time[4,48], a subset of specific proteins are temporally stable over many months and are well suited for non-invasive sampling. Our analysis of temporal stability of the urine proteome was restricted to prostate cancer patients that all have cISUP GG 1 tumors, similar serum PSA levels, and who did not have clinical upgrading over a period of 1–2 years. Follow-up studies utilizing larger, clinically heterogeneous cohorts will be needed to verify if the temporally stable proteins identified in low-grade patients generalize to other patients with other tumor grades. We found that the protein composition of urinary proteomes uniquely differs as a function of tumor grade, and when comparing prostate cancers to non-cancers. Consistent with data from other groups, prostate proteins such as PSA were elevated in the urine and urinary EVs of patients without cancer compared to those with prostate cancer, while serum PSA levels display the opposite trend[38,39,49]. This is likely due to the disruption of prostate cellular architecture with increasing tumor grade, which results in elevated levels of PSA in the bloodstream and consequently decreased levels of PSA in urine[38]. These differences in the secreted proteome between groups likely reflect altered signaling and intercellular communication by cancer cells or as a compensatory mechanism for altered tumor metabolism or cellular stress[50,51]. Therefore, we show that EV cargo is context-dependent and is associated with different cellular phenotypes, such as disease.

Further proteomic interrogation of urinary EVs in individuals of different ancestries, ages, and in different clinical scenarios[52] could offer new insights into genitourinary biology. However, the complexity of EV isolation by differential ultracentrifugation and the requirement for high starting material can be limiting[17]. Affinity-based capture of specific EV subpopulations using cell surface markers is appealing[53]. Intriguingly, uEVs appears to be characterized by cell surface markers different from other EV subpopulations[41]. Thus new protocols to rapidly isolate EVs from urine are urgently needed to maximize its utility for biomarker and other translational studies. In this study, we used differential ultracentrifugation, which is a less specific EV isolation approach compared to affinity-based capture or differential gradient ultracentrifugation which can isolate EV populations with higher specificity[54]. As such, the selectivity, protein composition, and protein abundance in EVs may differ depending on the choice of the EV isolation method. Future experiments will be needed to verify if these putative markers are able to capture similar EV subpopulations.

Since our unique dataset is one of the largest urine proteomics studies, we were able to leverage urinary fraction-specificity, frequency of detection in prostate tissues, and longitudinal stability to prioritize proteins as new biomarker candidates for prostatic diseases. The proteomics dataset generated in this study comprises one of the larger-sized discovery cohorts for urinary proteomics studies totaling >100 patients for prostate cancer biomarker discovery in urine, even for urinary EVs, and is well-powered to detect $\log_2$ fold changes in protein abundances of 1. However, while many of the proteomic differences between groups (e.g., cISUP GG) are concordant in independent cohorts, these differences are small. Further validation of the urinary signatures using robust quantitation by targeted proteomics assays with stable isotope labeled standards[52,55] in larger, racially diverse validation cohorts is needed. The full data resource presented

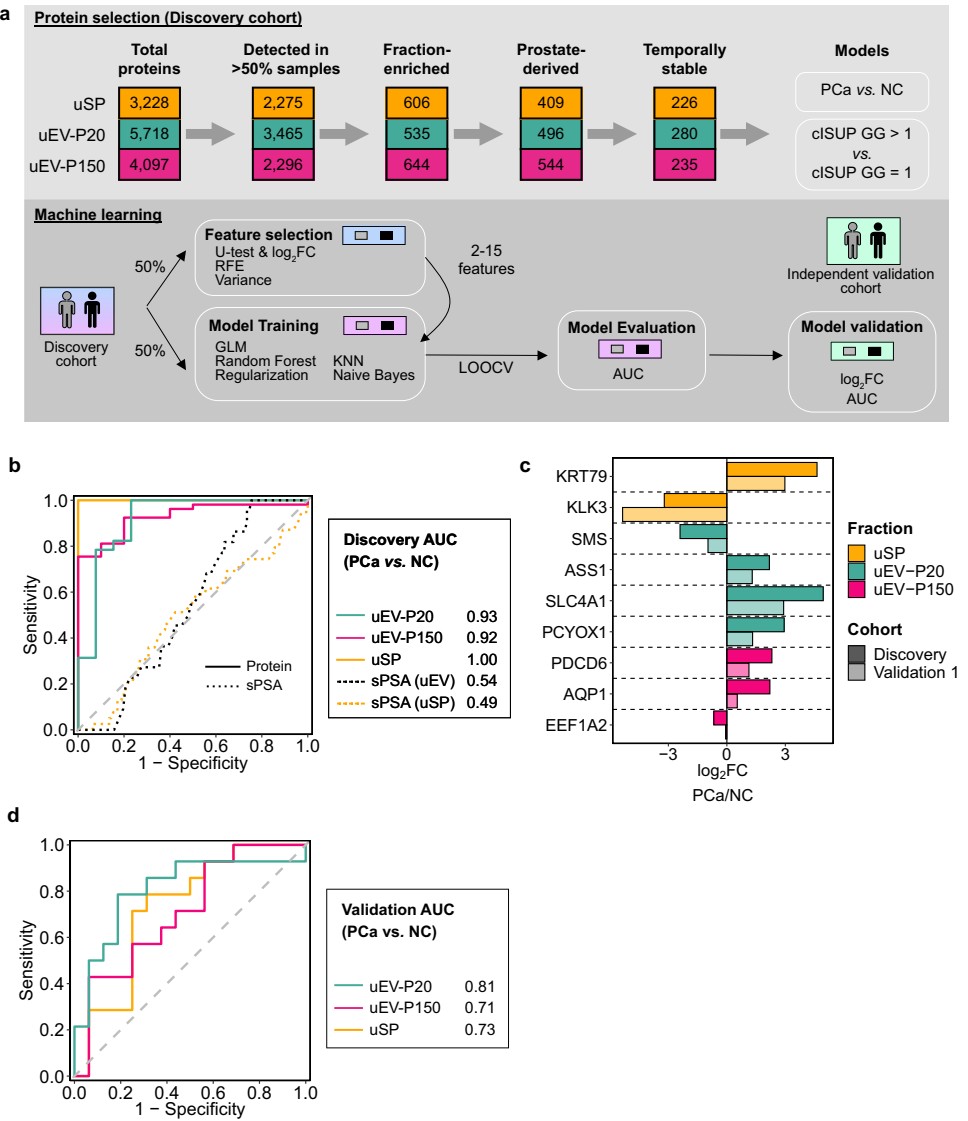

**Fig. 6 | Prioritization of urinary proteins for biomarker discovery. a** Strategy for urinary protein selection and building models to classify prostate cancers (PCa) vs. non-cancers (NC), and cISUP GG 1 vs. GG > 1. uSP: urinary soluble proteins; *U* test: two-sided Mann–Whitney *U* test; RFE: recursive feature elimination; GLM: generalized linear model; KNN: k-nearest neighbors; LOOCV: leave-one-out cross-validation; AUC: area under the receiver operator characteristic (ROC) curve. **b** ROC curves for multi-protein models (solid line) and serum PSA (dotted line) for classifying PCa vs. NC in the discovery cohort. (Patients: $uSP_{NC} = 39$, $uSP_{PCa} = 136$, uEV-P20$_{NC}$ = 22, uEV-P20$_{PCa}$ = 132, uEV-P150$_{NC}$ = 25, uEV-P150$_{PCa}$ = 131). **c** log$_2$ fold changes (log$_2$FC) of proteins in multi-protein models (**b**) in the discovery cohort (Patients: $uSP_{NC} = 39$, $uSP_{PCa} = 136$, uEV-P20$_{NC}$ = 22, uEV-P20$_{PCa}$ = 132, uEV-P150$_{NC}$ = 25, uEV-P150$_{PCa}$ = 131) and an independent prospective validation cohort (30 patients each for uSP, uEV-P20 and uEV-P150; PCa = 16, NC = 14). **d** ROC curves for multi-protein models and serum PSA in classifying PCa vs. NC in an independent validation cohort (30 patients each for uSP, uEV-P20, and uEV-P150; PCa = 16, NC = 14). Source data are provided as a Source Data file.

here is accessible via an interactive portal (http://kislingerlab.uhnres. utoronto.ca/ev/home/) to facilitate investigation into the urinary secreted and EV proteomes.

## Methods
### Human subjects
Samples were obtained from men following informed consent and use of Institutional Review Board-approved protocols at Eastern Virginia Medical School (EVMS, Norfolk, Virginia, USA, IRB# 06-12-FB-0343), Sunnybrook Health Sciences Centre (SHSC, Toronto, Ontario, Canada, Project #2457) and the Research Ethics Review Board at the University Health Network (UHN, Toronto, Ontario, Canada, 10-0159 and 19-5009). Men with benign prostatic conditions (non-cancer [NC]) (Supplementary Data 1) included individuals with elevated serum PSA (sPSA) levels and benign prostatic hyperplasia (BPH; 44 patients; median sPSA 6.3 ng mL$^{-1}$, range 1.7–11.9 ng mL$^{-1}$) or no diagnosed

prostate cancer on transrectal ultrasound-guided 12-core biopsy (Biopsy-negative; 20 patients; median sPSA 5.2 ng mL$^{-1}$, range 0.5–31.5 ng mL$^{-1}$). Selection criteria for men with benign prostatic conditions included a diagnostic sPSA level <20 ng mL$^{-1}$ and post-surgery sPSA level <0.1 ng mL$^{-1}$ to exclude highly metastatic men. Other clinical details are detailed in Supplementary Data 1.

### Urine collection
The first 15 mL of first-catch urine collected post-digital rectal exam (DRE) (post-DRE urine) was collected by performing a gentle massage of the prostate gland during DRE prior to biopsy[2]. For the DRE cohort (Supplementary Data 1), which comprised of ten men with clinical ISUP GG 1 tumors, mid-stream urine was collected an hour before the DRE massage (pre-DRE urine). Matched post-DRE urine was also collected for these ten men. The longitudinal cohort comprised of five men with cISUP GG 1 tumors who are on active surveillance and did not upgrade

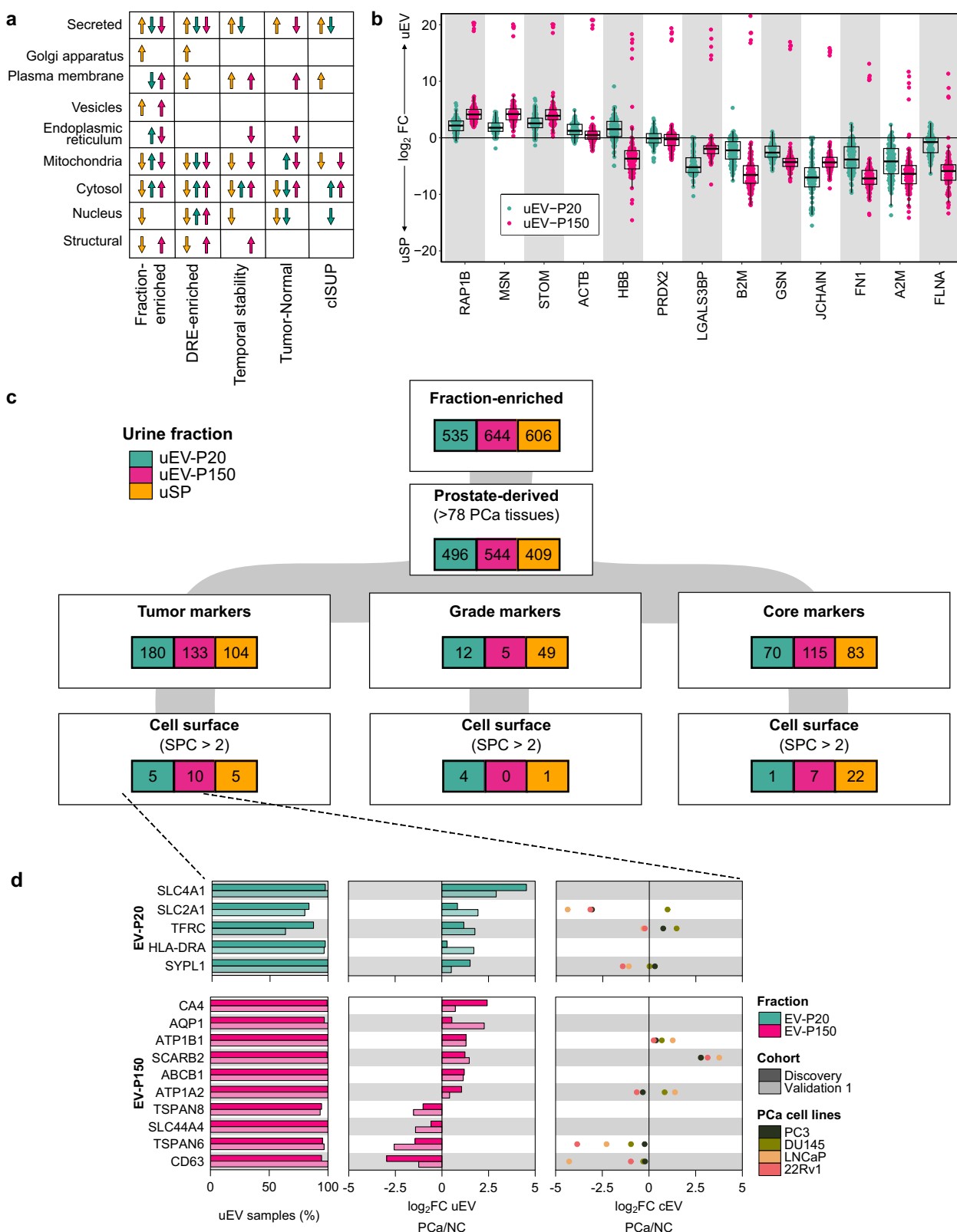

in the period of 12–16 months after their first DRE (Supplementary Data 1). Serial post-DRE urine was collected for each patient at three-time points. Each time point was 3–12 months apart. For assessing the reproducibility of uEV isolation in prostate cancer patients, the first 50 mL of first-catch post-DRE urine was collected from three men with cISUP GG 1 tumors. For assessing the reproducibility of uEV isolation in

men without prostate cancer, we pooled post-DRE urines from 10 men with benign prostatic hyperplasia and pooled post-DRE urines from 10 men with elevated serum PSA but no prostate cancer detected on needle biopsy. Pre- and post-DRE urine was centrifuged at 2000 × *g* for 15 min at 4 °C to pellet cellular debris, and the resulting urine supernatant was stored at −80 °C.

**Fig. 7 | EV cargo is context-dependent. a** Proteins differentially abundant in each condition and urine fraction by compartment. **b** Differences in protein abundance of extracellular vesicle markers from ref.[41] in matched uEV-P20 (green) and uEV-P150 (pink) relative to uSP. $n = 96$ patients. Boxplots are shown with the line indicating the sample median, the box indicating the 25th and 75th percentiles, and the whiskers indicating $\pm 1.5 \times$ IQR. **c** Fraction-specific proteins were determined by differential protein abundance and frequency of detection (Fig. 2j). Proteins in each subset were filtered based on the following criteria. Tumor markers: $|\log_2 FC|_{PCa/NC} > 1$ and FDR$_{PCa/NC} < 0.05$; Grade markers: $|\log_2 FC|_{cISUP>1/cISUP=1} > 1$ and unadjusted $P_{cISUP>1/cISUP=1} < 0.05$; Core markers: $|\log_2 FC|_{PCa/NC} < 1$ and FDR$_{PCa/NC} \geq 0.05$ and $|\log_2 FC|_{cISUP>1/cISUP=1} > 1$ and unadjusted $P_{cISUP>1/cISUP=1} \geq 0.05$ and >90% of samples and <25% least variant proteins. Tumor, grade, and core markers were further filtered to select cell surface markers (SPC[42] >2). PCa: Prostate cancer; NC: Non-cancer; SPC: Surface prediction consensus[42]. $P$-values from a two-sided Mann–Whitney $U$ test. **d** Tumor markers with predicted cell surface localization from (**c**). Proteins are annotated with the frequency of detection in each fraction (left panel) and differential abundance in PCa vs. NC in uEV (middle panel) and cEV (right panel). Non-cancer (NC) cEVs from RWPE1 cell line. Source data are provided as a Source Data file.

## Cell lines

Commercial human prostate cell lines DU145 (ATCC #HTB-81), PC3 (ATCC #CRL-1435), 22Rv1 (ATCC #CRL-2505), LNCaP (ATCC #CRL-1740), and RWPE1 (ATCC #CRL-3607) were a gift from Dr. Stanley Liu, Sunnybrook Health Sciences Centre, Toronto, Ontario, Canada. All cell lines are immortalized cell lines from males. Cell line identity was confirmed by short tandem repeat testing. Mycoplasma negativity was confirmed using the Universal Mycoplasma Testing Kit (ATCC). Cells were seeded in two T500 Nunc™ TripleFlasks™ (total area = 1000 cm²) with 100 mL of media or ten 15 cm plates (total area = 1480 cm²) in 20 mL of media each and cultured in a 37 °C incubator with 5% CO₂. RPMI media (Gibco) supplemented with 10% fetal bovine serum and 1% penicillin-streptomycin-glutamine (PSG) was used for the prostate cancer cell lines (DU145, PC3, 22Rv1 and LNCaP) and Keratinocyte-serum free media supplemented with 0.05 mg mL⁻¹ bovine pituitary extract, 5 ng mL⁻¹ epidermal growth factor, and 1% Penicillin-Streptomycin-Glutamine was used for the RWPE1 cell line.

## EV isolation from urine

Urinary extracellular vesicles (uEV) were isolated by differential ultracentrifugation[17]. Briefly, 14 mL of frozen urine supernatant was thawed at 4 °C, then diluted to a volume of 35 mL with isotonic buffer (250 mM sucrose, 10 mM HEPES, 1 mM EDTA, pH 7.4). The urine was centrifuged at 20,000 × g for 30 min at 4 °C (k-factor 1790) in an Optima XPN-80 ultracentrifuge (Beckman Coulter) equipped with a SW32Ti swinging bucket rotor (R$_{min}$ 67, R$_{max}$ 153, Beckman Coulter) to pellet EVs. The 20,000 × g pellet (P20) was treated with 500 mM of dithiothreitol (DTT) at 37 °C for 30 min to reduce the uromodulin network and centrifuged a second time at 20,000 × g for 30 min at room temperature. The P20 pellet was resuspended in 1 mL of cold PBS and centrifuged at 18,210 × g for 30 min (Eppendorf Centrifuge 5430 R, FA-45-48-11 rotor, k-factor 198). The supernatant from the first and second centrifugation steps were combined and centrifuged at 150,000 × g for 2 h at 4 °C (SW32Ti swinging bucket rotor, k-factor 239) in an ultracentrifuge to pellet EVs. The 150,000 × g pellet (P150) was resuspended in a high pH buffer and then passed twice through a 0.22 μm filter. Samples were centrifuged again at 150,000 × g for 2 h at 4 °C to pellet uEV-P150. The P20 and P150 pellets containing uEV-P20 and uEV-P150, respectively, were resuspended in 100 μL of 50% *2,2,2*-trifluoroethanol (Sigma–Aldrich) in PBS, flash-frozen in liquid nitrogen, and stored at −80 °C until proteomics analysis.

## EV isolation from cell culture-conditioned media

All cell lines were grown to 70–80% confluency then washed three times with phosphate-buffered saline (PBS) and serum-starved for 48 h prior collection of conditioned media. The cell line conditioned media containing EVs (cEV) was collected and centrifuged at 500 × g for 10 min then 2000 × g at 4 °C for 30 min to clear cell debris. The supernatant was concentrated to a volume of 4–5 mL (if using EVs for biophysical studies) or to a volume of 20–30 mL (if using EVs for proteomics) in a 100 kDa MWCO ultrafiltration concentrator (Millipore). EVs were isolated from conditioned media by differential ultracentrifugation in a SW32Ti swinging bucket rotor as described above. Conditioned media was topped off with PBS as required. Unlike the EV isolation protocol for urine described above, the first P20 pellet was not treated with DTT as cell line conditioned media is not expected to contain uromodulin. cEVs were collected from the 20,000 × g (cEV-P20) and 150,000 × g (cEV-P150) pellets.

## Urine proteomics

Proteomic profiles of the soluble protein fraction were generated from 250 μL of urine supernatant (following 2000 × g centrifugation). Urine was prepared for proteomics using the MStern protocol[56]. For each sample, 2 pmol of *Saccharomyces cerevisiae* invertase was added as a sample preparation control. Proteins in each sample were reduced with 5 mM DTT and incubated for 30 min at 60 °C. To prevent re-formation of disulfide bonds, 25 mM iodoacetamide was added and samples were incubated at room temperature for 30 min in the dark. The liquid in the following steps was passed through the MStern wells using vacuum suction unless otherwise stated. The polyvinylidene fluoride membrane (Millipore Sigma, MSIP4510) was equilibrated with 50 μL of 70% ethanol, then washed twice with 100 mM ammonium bicarbonate (ABC). Samples were added to the wells and passed through the membrane by vacuum suction. Each well was washed twice with 100 μL of ABC to remove salts, then proteins were digested with 1 μg of mass spectrometry grade Trypsin/Lys-C enzyme mix (Promega) in 50 μL of digestion buffer (100 mM ABC, pH 8.0, 1 mM CaCl₂, 5% acetonitrile). To ensure that the proteins are in contact with the digestion buffer, the digestion buffer was passed through the membrane by centrifugation, and the flow-through was reapplied on top of the membrane. Protein digestion was performed at 37 °C for four hours. Samples were resuspended in the well by gentle pipetting every two hours. Peptides were collected by centrifugation, and remnant membrane-bound peptides were eluted with 50 μL of 50% acetonitrile and combined with the previous flow-through. Samples were dried in a SpeedVac vacuum concentrator (Thermo). Dried peptides were resuspended in 0.1% trifluoroacetic acid in water and desalted using homemade solid phase extraction stage tips containing 3 plugs of 3M™ Empore™ C18 membrane[57]. Peptides were quantified by NanoDrop (Thermo Scientific). 2 μg of peptides were loaded on the column.

## Extracellular vesicle proteomics

EVs or whole-cell lysates in 50% *2,2,2*-trifluoroethanol (Sigma–Aldrich) were lysed by freeze-thaw, then incubated at 60 °C for 1 h to extract proteins. Then, proteins were reduced with 5 mM of DTT, alkylated with 25 mM of iodoacetamide, and digested overnight at 37 °C with a 2 μg Trypsin/Lys-C enzyme mix (Promega). The next day, the enzymatic digest was quenched with 1% formic acid, and samples were desalted with homemade C18 StageTips (see above) prior to LC-MS analysis. iRT peptide standards (Biognosys) were spiked into reconstituted peptides at a 1:100 dilution according to the manufacturer's instructions.

## Mass spectrometry and data processing

Peptides were separated on a 50 cm C18 reverse phase EASY-Spray LC column (Thermo ES803) with trap column (Acclaim™ PepMap™ 100 C18) interfaced with an EASY-nanoLC 1000 system over a 2 h gradient

(EVs and urine) or 4 h gradient (whole-cell lysates). Mass spectrometry was performed on a Q-Exactive HF, Orbitrap Fusion Tribrid or Orbitrap Fusion Lumos mass spectrometer coupled to an EASY-Spray ESI source (Thermo Scientific). Mass spectrometry data acquisition parameters and replication for each cohort are listed in Supplementary Data 5. All datasets were acquired in data-dependent acquisition mode. Raw files for each urine fraction and cohort were searched separately in MaxQuant[58] (v.1.5.8.3 for uSP samples and v.1.6.2.3 for uEV samples) at a single site using a UniProt human protein sequence database (complete human proteome with isoforms). For cohorts with samples processed or acquired in replicates, protein intensities were combined in MaxQuant. Searches were performed with trypsin cleavage at lysine and arginine, maximum of two missed cleavages, peptide length 7–25 amino acids, and carbamidomethylation of cysteine as a fixed modification. Variable modifications were set as oxidation of methionine and acetylation of the protein N-terminus. The false discovery rate for the target-decoy search was set to 1% for protein and peptide. Peptide detection was performed with an initial precursor and fragment mass deviation threshold of 10 and 20 parts per million respectively. Intensity-based absolute quantification (iBAQ), label-free quantitation, and match between runs (matching and alignment time windows set as 0.7 and 20 min respectively) were enabled. The *peptides.txt* output files from each MaxQuant search were parsed into an in-house database for protein grouping[59]. Protein abundances (gene-centric) were determined from peptide abundances using the iBAQ algorithm[60] (Supplementary Data 2). Reverse hits (false positives from target-decoy search) were removed, and proteins detected with two or more peptides were carried forward. Raw iBAQ intensities were normalized using median normalization. Median-normalized values were used for all analyses unless stated otherwise. All further data analysis was performed in the R statistical environment (v.4.2.1).

### EV isolation for biophysical studies
EVs were isolated from urines or cell line conditioned media for nanoparticle tracking analysis (NTA) and transmission electron microscopy (TEM) as described above, with the following changes. 5 mL of fluid was used for EV isolation using a SW55 Ti swinging bucket rotor ($R_{min}$ 61, $R_{max}$ 109, Beckman Coulter). To keep the k-factor consistent with the SW32Ti rotor at each centrifugation step and taking into account the time needed for the rotor to achieve its desired speed (approximately 5 min), centrifugation time were reduced to 20 min and 1 h for the $20,000 \times g$ (k-factor 699) and $150,000 \times g$ (k-factor 120) centrifugation steps, respectively. EV pellets were resuspended in 100–200 μL of cold, 0.22 μm filtered PBS, and stored at 4 °C for no more than 16 h prior to NTA or TEM analysis.

### Transmission electron microscopy
TEM was performed at SickKids Nanoscale Biomedical Imaging Facility and Princess Margaret Cancer Centre. Samples were deposited on formvar carbon-coated grids, washed once with water, and stained twice with uranyl acetate. Images were acquired on a Tecnai 20, Hitachi HT7800, and Talos™ F200X G2 transmission electron microscope. Images were processed with ImageJ (v.1.53t) for visualization.

### Nanoparticle tracking analysis
NTA for 34 men (Supplementary Data 1) with uEV-P20 and uEV-P150 was performed using a NanoSight LM10 system configured with a 405 nm laser and a high-sensitivity sCMOS camera. Camera settings were as follows: screen gain 3.0; camera level 11; 25 frames per second; slider gain 146. Each sample was diluted in particle-free PBS and introduced manually. Analysis was performed with NTA software (v.3.1 build 3.1.46). One technical replicate was captured per sample. For each replicate, three 30 s videos were captured, with approximately 20–200 particles in the field of view for each measurement. Ambient temperature was set at 22 °C.

NTA for the DRE cohort (pre-DRE vs. post-DRE urine from two matched men) and cEVs (cEV-P20 and cEV-P150) was performed using a NanoSight NS300 system (Malvern) configured with a 405 nm laser and a high-sensitivity sCMOS camera. Each sample was diluted in 0.22 μm filtered PBS and introduced with a syringe pump at 60 μL min⁻¹. Analysis was performed with NTA software (v.3.4). For every sample, two to four technical replicates were captured. For each technical replicate, three 30 s videos were captured, with approximately 20–200 particles in the field of view for each measurement. Ambient temperature was set at 22 °C.

Raw data files ("*filename*-ExperimentSummary.csv") were parsed as follows for quantification and statistical analysis. Raw particle counts for each size bin were corrected for dilution factors, and then grouped to biological replicates. Each data point represents the mean of all measurements for each biological replicate (Supplementary Data 1). For cEVs, an experimental replicate is defined as cEVs isolated from the same cell line at different passages. For uEVs, a biological replicate is defined as uEVs isolated from a specific biofluid (pre-DRE or post-DRE urine) from one individual. For visualizing particle concentration vs. size distribution for each replicate (Supplementary Fig. 1d, 2c, and S2d) particle concentration was scaled with min-max [0,1] normalization with formula Eq. (1).

$$x' = (x - \min(x))/(\max(x) - \min(x)) \tag{1}$$

### Quantification and statistical analysis
Where appropriate, quantitative analyses are described in the relevant sections of the Methods. Unless stated otherwise, bioinformatic and statistical analyses and plotting were performed using R (v.4.2.1). Data were visualized using R packages *BoutrosLab.plotting.general* (v.7.0.3), *ggplot2* (v.3.4.0), *ggbeeswarm* (v.0.6.0) *ggpubr* (v.0.4.0), and *ComplexHeatmap* (v.2.12.1). Qualitative variables were compared by Fisher Exact Test, and quantitative variables by two-sided Mann–Whitney *U* test for unpaired comparisons (*wilcox.test*), two-sided Wilcoxon signed-rank test for pairwise comparisons (*wilcox.test*) and the Kruskal–Wallis test for multiple group comparisons. $\log_2$ fold change ($\log_2$FC) is calculated from the difference in medians. The specific statistical tests used are indicated in the figure legends. Multiple test *P*-values were adjusted using Benjamini–Hochberg method for independent tests unless stated otherwise. Correlation coefficients were determined by the Spearman method (*cor.test*). *P*-values for Spearman's correlation were computed by asymptotic t approximation using an Edgeworth series. Statistical significance was set at *P*-value < 0.05. Missing values in protein-level data were replaced by random numbers drawn from the lower tail of the Gaussian distribution 1.8 standard deviations from the mean (width = 0.2 standard deviations) unless stated otherwise[61].

### Tissue specificity in DRE urines
To determine if DRE enriches for prostate tumor-derived proteins, proteomic profiles for the soluble protein (uSP) and uEV fractions were generated from urines collected pre- and post-DRE. Paired Student's t-tests were used to identify differentially abundant proteins in pre- and post-DRE. To determine which proteins are anticipated to be derived from prostate tumors, proteins in this study were annotated based on detection in more than 10 tissue samples of two prostate cancer tissue proteomic datasets: Sinha et al.[18] (76 men with prostate cancer; 76 tumor samples) and Khoo et al.[19] (40 men with Prostate cancer; 81 samples [41 tumor and 40 NAT]); 7438 proteins. To determine which proteins had elevated expression in human prostate tissue, the Human Protein Atlas (v.21.1, updated 2022-05-31) Human Tissue-Specific Proteome[20] from the prostate was used to annotate proteins detected in this study. Proteins were included if they belonged to the

'Tissue enriched', 'Group enriched' or Tissue enhanced' categories for the tissue of interest, totaling 127 genes for prostate tissue.

## Sample type correlations

Median protein abundances across samples were used for all comparisons using Spearman's rank correlation (*cor.test*). *P*-values for Spearman's correlation were computed by asymptotic t approximation using an Edgeworth series. Samples: uEV-P20 = 146, uEV-P150 = 148, Tissue = 157, uSP = 175. Proteins used for each comparison: uEV-P20 vs. uEV-P150 = 3593; uSP vs. uEV-P20 = 2839; uSP vs. uEV-P150 = 2309; Tissue vs. uSP = 2626; Tissue vs. uEV-P20 = 4735; Tissue vs. uEV-P150 = 3410.

## Identifying and annotating sample type-enriched proteins

To identify proteins enriched in uSP, uEV-P20 or uEV-P150 fractions, we considered both unique and differentially abundant proteins in each fraction. The set of fraction-unique proteins were defined as proteins detected in >90% of samples of one fraction and detected in less than 10% of samples of the other two fractions. Only samples with matched uSP, uEV-P20, and uEV-P150 fractions were used (288 samples from 96 patients). To identify differentially abundant proteins, proteins present in >20% of samples were used (Proteins: uSP = 2909; uEV-P20 = 4841, uEV-P150 = 3389). A two-tailed, paired two-sided Wilcoxon signed-rank test was used to compare protein abundance in uSP vs. uEV-P20 (2430 shared proteins), uSP vs. uEV-P150 (1950 shared proteins), and uEV-P20 vs. uEV-P150 (2939 shared proteins). Proteins were considered fraction-elevated if they were differentially abundant in both comparisons (Fig. 2i). For example, proteins were considered uSP-unique if they were detected in >90% of uSP samples, in <10% of uEV-P20 samples, and in <10% of uEV-P150 samples (Fig. 2h). Proteins were considered 'uSP-elevated' if they were more abundant in uSP vs. uEV-P20 and uSP vs. uEV-P150 comparisons (FDR < 0.05 and $|\log_2 FC| > 0$, intersect: 516 uSP-enriched proteins) (Fig. 2i). This process resulted in a total of 606 uSP-enriched proteins (Fig. 2j).

Fraction-enriched proteins were annotated with subcellular localization information from nine main categories (Secreted, Vesicles, Plasma membrane, Mitochondria, Cytosol, Nuclear Membrane, Nucleoplasm, Nucleoli, and Golgi apparatus) from Human Protein Atlas' Subcellular location data[62] (v.22.0, proteinatlas.org). "Nuclear membrane", "Nucleoplasm" and "Nucleoli" categories were collapsed into one category called "Nucleus". Fisher's Exact Test was used to test for over- or under-representation in each category for each fraction. The magnitude of the enrichment was estimated using the odds ratio (*epitools* v.0.5–10.1), with the union of proteins detected in fluids (uSP, uEV-P20, and uEV-P150; 6540 proteins) used as a custom background.

## Sample type tissue enrichment scores

To score samples based on their tissue content, Gene Set Variation Analysis (v.1.44.5)[63] was used to score samples using two custom gene signature sets – prostate and non-prostate (bladder + kidney). For the signature gene sets, proteins that were enriched in each tissue type in the Genotype-Tissue Expression (GTEx) Project Bulk Tissue RNA-Seq dataset[25] (V8; retrieved 2017-06-05; $n_{Prostate}$ = 245 samples; $n_{Kidney}$ = 89 samples; $n_{Bladder}$ = 21 samples) and The Cancer Genome Atlas (TCGA) normal tissue adjacent to tumor from males[26–28] (NAT; TCGA v.2016_01_28; $n_{PRAD}$ = 52 samples, 19,821 genes; $n_{BLCA}$ = 10 samples, 18,951 genes; $n_{KIRP}$ = 22 samples, 19,518 genes; $n_{KIRC}$ = 52 samples, 19,667 genes). For KIRP and KIRC NAT samples, we selected duplicated samples that had Spearman's ρ > 0.99 and took the mean intensity, leaving a total of 71 KIRP/KIRC samples (19,829 genes). For each dataset, tissue-enriched genes were determined using two-sided Mann–Whitney *U* tests of each tissue of interest vs. the other two tissue types (i.e., GTEx$_{Prostate}$ vs. GTEx$_{Bladder + Kidney}$; $\log_2 FC > 3$ and FDR < 0.05). For the gene set signature, concordant genes in GTEx[25]

and TCGA NAT[26–28] were selected (Genes: prostate = 46, bladder or kidney = 41).

## Pathway analysis – sample type comparison

Proteins detected in >10 samples of each sample type (Proteins: tissue = 7438, uSP = 3150, uEV-P20 = 5462, uEV-P150 = 3878) were used for pathway analysis using *gprofiler2* (v.0.2.1) and *g:GOSt()* against Gene Ontology:Cellular Component gene sets. Pathway enrichment was performed using default parameters (*organism* = "hsapiens", *significant* = TRUE, *user_threshold* = 0.05, *correction_method* = "g_SCS", *custom_bg* = NULL, *sources* = "GOCC"). *P*-values were determined using Fisher's one-tailed test and adjusted for multiple corrections using the g_SCS method[64]. Significantly enriched pathways were visualized using EnrichmentMap (v.3.3.4) in Cytoscape (v.3.9.1).

## Cell line proteomics data

Cells and EVs were collected from three separate passages for each cell line, termed experimental replicates, for proteomics. For each cell line and sample type, only proteins that were present in a minimum of two of three replicates were carried forward for analyses.

## Identifying temporally stable proteins

We generated proteomic profiles of uSP, uEV-P20, and uEV-P150 fractions from post-DRE urine from a longitudinal cohort composed of five patients. These patients had cISUP GG 1 tumors and were on active surveillance (Supplementary Data 1). None of the men upgraded in the time that the urines were collected. For statistical analysis of the longitudinal cohort, only reproducibly detected proteins were included for analysis. For each urine fraction, we selected proteins that were detected in at least two-time points for each patient and detected in at least two patients. This resulted in a total of 1664 uSP proteins, 3365 uEV-P20 proteins, and 1990 uEV-P150 proteins. The similarity of intra-patient and inter-patient proteomes were determined using Spearman's correlation, calculated using the *cor()* function (*use* = "pairwise.complete.obs") in *stats* package in R (v.4.2.1).

Intra- and inter-individual variance in protein intensities was assessed using linear mixed-effects regression using the *lme4* package (v.1.1–31), and the intraclass correlation coefficient (ICC) was measured, which represents the proportion of inter-individual variance relative to the total intra- and inter-individual variance explained by a model[37]. Proteins for which a model cannot be fitted due to random effect variances of close to zero. This resulted in a total of 1664 uSP proteins, 3365 uEV-P20 proteins, and 1990 uEV-P150 proteins with estimated ICC values.

## Prostate cancer vs. non-cancer comparisons

Proteins detected in >50% of each sample type were considered for differential abundance analysis. This resulted in a total of 2156 uSP proteins, 3431 uEV-P20 proteins, and 2255 uEV-P150 proteins. Two-sided Mann–Whitney *U* test was used for comparisons. For each sample type, gene set enrichment analysis (GSEA v.4.3.2) was performed on a pre-ranked list of proteins based on differential abundance ($\log_2$ fold change) in prostate cancers and non-cancers. Enrichment analysis was performed against the human MSigDB Hallmarks gene set (v.2022.1) with gene set sizes from 25–500 and 1000 permutations.

## Feature selection and machine learning

To generate predictors that distinguishes men with prostate cancers and non-cancers (NC), proteomics data from uSP, uEV-P20, and uEV-P150 fractions were trained separately. Patients: uSP$_{NC}$ = 39, uSP$_{PCa}$ = 136, uEV-P20$_{NC}$ = 22, uEV-P20$_{PCa}$ = 132, uEV-P150$_{NC}$ = 25, uEV-P150$_{PCa}$ = 131. The same cohorts were used to generate predictors that distinguishes men with cISUP GG 1 from men with cISUP GG >1 (uSP: 50 GG 1 vs. 61 GG >1; uEV-P20: 41 GG 1 vs. 63 GG >1; uEV-P150: 40 GG 1

vs. 63 GG > 1). To develop predictive models, all datasets were divided into two groups: feature selection (50% of the dataset) and training (50% of the dataset). Within each urinary fraction, proteins that were detected in more than 50% of all samples and temporally stable (intraclass coefficient > 0.4 from serial post-DRE urine collected from three cISUP GG 1 patients at three-time points) were passed into feature selection. Three methods were used to select the top 2–15 features within each dataset. For each feature, $\log_2$FC protein abundance was calculated and the significance level was assessed using a two-sided Mann–Whitney $U$ test (*wilcox.test*). Features with the small*est* $P$-values were selected as the first set of top features. Features with the highest $\log_2$FC and $P$-value < 0.001 were also selected as the second set of top features. Ten times repeated five-fold cross-validated (*rfeControl*) was applied to get the third set of top features. Seven machine-learning algorithms were applied to the top features in the biomarker identification, including generalized linear models, random forest, k-nearest neighbor classification, naïve bayes, ridge, lasso, and elastic-net-regularized generalized linear model. Receiver operating characteristic (ROC) analysis with leave-one-out cross-validation was used to evaluate model performance with the use of '*pROC*' package (v.1.18.0). Models with the highest area under the ROC were chosen and were fit to the entire dataset to get the final model. Machine-learning algorithms were performed using the *caret* package (v.6.0.91) in R (v.3.6.1).

### Identifying context-driven urine proteins

For each urine fraction (uEV-P20, uEV-P150, and uSP), we sought to identify proteins that were distinct to each urine fraction, prostate-derived, and indicative of disease state (prostate cancer vs. non-cancer or cISUP GG > 1 vs. cISUP GG 1). We also sought to identify fraction-specific, prostate-derived proteins that were stable in protein abundance, independent of disease state (i.e., core proteome). Of the fraction-enriched proteins previously identified in Fig. 2j (uEV-P20: 535 proteins, uEV-P150: 644 proteins, uSP: 606 proteins), we filtered for proteins that were detected in >50% of prostate tissue samples (i.e., in more than 78 prostate tissues)[18,19]. We called these proteins fraction-enriched, prostate-derived proteins (uEV-P20: 496 proteins, uEV-P150: 544 proteins, uSP: 409 proteins). From this set of genes, we identified proteins that were stable across disease conditions, making up the core proteome – detected in >90% of samples in each fraction, not differentially abundant in prostate cancers vs. non-cancers ( $|\log_2$FC $|$ <0.5), not differentially abundant in cISUP GG > 1 vs. cISUP GG 1 ( $|\log_2$FC $|$ <0.5), and in the bottom 25% least variable proteins. This resulted in 70 uEV-P20 proteins, 115 uEV-P150 proteins, and 83 uSP proteins. We also identified proteins that were differentially abundant in prostate cancers vs. non-cancers (i.e., "Tumor markers"; $|\log_2$FC $| > 1$ and FDR < 0.05) and in cISUP GG > 1 vs. cISUP GG 1 (i.e., "Grade markers"; $|\log_2$FC $| > 1$ and unadjusted $P$-value < 0.05). From each of these groups – Core, Tumor, and Grade markers – we identified proteins with predicted cell surface localization (Surface Prediction Consensus [SPC] score[42] > 2) that could serve as potential markers for each of these groups.

### Reporting summary

Further information on research design is available in the Nature Portfolio Reporting Summary linked to this article.

## Data availability

The mass spectrometry raw data generated in this study have been deposited in the UCSD MassIVE database under accession code MSV000092061 [https://massive.ucsd.edu/ProteoSAFe/dataset.jsp?task=28fc0fb96b1f4a17870b85c5d1789f6e]. The processed proteomics data are provided in Supplementary Data 2. All relevant data for the EV experiments have been submitted to the EV-TRACK knowledgebase[65] (EV-TRACK ID: EV230578). An accompanying website for this study provides an interactive browser for the interrogation of multiple datasets and clinical cohorts, available at http://kislingerlab.uhnres.utoronto.ca/ev/home/. Published proteomics datasets used in this study[17–19,22] can be accessed at UCSD's MassIVE database under accession codes MSV000088000 and MSV000081552 or PRIDE repository PXD015289 and PXD026974. Source data are provided with this paper.

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

## Acknowledgements

We thank the Nanoscale Biomedical Imaging Facility at SickKids and Mohammad Mazhab-Jafari at Princess Margaret Cancer Centre for acquiring the negative stain transmission electron microscopy images. We would also like to thank the SickKids Structural & Biophysical Core Facility for allowing the use of their instruments for nanoparticle tracking analysis. The Genotype-Tissue Expression (GTEx) Project was supported by the Common Fund of the Office of the Director of the National Institutes of Health, and by NCI, NHGRI, NHLBI, NIDA, NIMH, and NINDS. The results here are in whole or part based upon data generated by the TCGA Research Network: https://www.cancer.gov/tcga. A.K., M.G., and L.Y.L. are supported by the Ontario Graduate Scholarship and Ontario Student Opportunity Trust Fund Awards. L.Y.L. is supported by a Vanier Canada Graduate Scholarship. This study was supported by the National Institutes of Health through awards U01CA214194, U2CCA271894, U24CA248265, R01CA272678, P20CA252717 and UM1TR004360. This work was supported by a Prostate Cancer Foundation Special Challenge Award to P.C.B. (20CHAS01) made possible by the generosity of Mr. Larry Ruvo. This study was supported by Canadian Institutes of Health Research Project Grants to T.K. (PJT156357) and S.K.L. (PJT162384). T.K. was supported through the Canadian Research Chair program. T.K., S.K.L., and D.V. also received support through a Prostate Cancer Canada Discovery Grant (D2019-2113).

## Author contributions

A.K., M.G., A.M., Al.K., S.N., B.P.M., R.S.L., L.Y., M.D., D.V., S.K.L., O.J.S., and J.O.N. contributed to data acquisition. A.K. and A.M. acquired the mass spectrometry data. A.K., Z.Q., M.G., L.Y.L., and V.I. analyzed the data. S.K.L., O.J.S., J.O.N., P.C.B., and T.K. supervised the study. A.K., M.G., Z.Q., M.W., L.Y.L., T.K., and P.C.B. wrote the manuscript, which all other authors edited and approved.

## Competing interests

P.C.B. declares the following competing interests: P.C.B. sits on the Scientific Advisory Boards of Sage Bionetworks, Intersect Diagnostics Inc., and BioSymetrics Inc. The remaining authors declare no competing interests.
