## [Peer Review File · Nature Communications]

Prostate Cancer Reshapes the Secreted and Extracellular Vesicle
Urinary ProteomesREVIEWER COMMENTS

Reviewer #1 (Remarks to the Author):

In this well-written manuscript, the authors generated comprehensive proteomic profiles of urine (urine soluble proteins (uSP) and urinary extracellular vesicle (uEV) proteins) from a mixed cohort of 190 men, including a subset with prostate cancer. They first show in matched pre- and post-digital rectal examination (DRE) urine from 10 men with low grade prostate cancer that DRE increases abundance of prostate-derived proteins in all urine fractions (uSP, uEV-P20, and uEV-P150) without influencing EV biophysical characteristics. They then analyze proteomes of post-DRE urine from 190 men, of whom 64 had no prostate cancer and 126 had untreated prostate cancer (54 with GG1, 20 GG2, 26 GG3, 11 GG4, 15 GG5). They characterize the protein cargo in different urine fractions and find that urine EVs are more reflective of the prostate tissue proteome, in particular uEV-P20. When compared with prostate cancer cell line-derived EVs, urine EVs have similar biophysical characteristics, but have protein composition more reflective of prostate tissues. Next they study a longitudinal cohort of 5 prostate cancer patients and identify EV proteins that are stable over time. They then compare urine proteomes of patients with and without prostate cancer, and then use machine learning to create classifiers to distinguish cancer from non-cancer, and to distinguish GG1 (low grade) from GG>1 (higher grade) cancer. Finally, they performed bioinformatic analyses to identify optimal protein markers that potentially could be used for affinity-based isolation of specific actionable EV subpopulations.

Overall this is a comprehensive analysis of urine EV proteomics in a large cohort of men with or without prostate cancer, with potential clinical implications for the development of minimally invasive diagnostic and active surveillance tools. The proteomics of urine EVs and their potential clinical utility has been studied extensively in the past with multiple publications in the literature, but this work is significant for the size of cohort and the depth of the proteomic analysis. The development of urine proteomic classifier models is especially interesting, although a separate validation cohort is missing. Additional comments are as follows:

1. Line 127-128 and Ext. Data Fig 2C: The text states that "uEV biophysics ... were largely independent of disease status...", but Ext. Data Fig. 2C shows that GG3 has more particles than the other grade groups (and significantly more than GG4-5). This might be a spurious finding due to low numbers of patients in each group, but the authors should address this in the text.
2. Line 132 and Ext. Data Fig. 3A: Although the majority of urine EV proteins identified in this study overlapped with those identified in previous studies, it appears some proteins were unique to this study. Were these newly identified urine EV proteins of any biological interest? Were any of them uniquely found only in certain patient subsets such as GG4 and GG5 cancers?
3. Fig. 5A-B: Although the longitudinal evaluation and protein variance analysis over time is interesting, this analysis is very limited given that only 5 patients are included.
4. Lines 212-215 and Figs. 5C-E: The finding that PSA (KLK3) protein abundance is consistently reduced in all fractions of urine from prostate cancer patients compared to men without cancer, despite being increased in serum and tumor, is puzzling. The authors should provide an explanation for this finding in their Discussion.
5. The urine proteomic classifiers (Figs. 5G-H) for distinguishing cancer vs. non-cancer and GG1 vs. GG>1 are promising, but they have been trained and evaluated in the same patient cohort using leave-one-out-cross validation. It will be necessary to validate these classifier models in separate independent patient cohorts.
6. It is interesting that among the various urine proteomic classifiers, the urine soluble protein (uSP) based models actually perform very well with a high AUC of 0.94 for cancer vs. non-cancer, and performs even better than the urine EV protein-based classifiers for GG1 vs. GG>1 (AUC 0.86 for uSP, compared to 0.79 for uEV-P20 and 0.67 for uEV-P150). This could be used to argue that EVs don't need to be isolated from urine at all, and that a simple urine test based just on just soluble proteins could be more useful as a biomarker as opposed to urine EVs.
7. Fig. 4: Can the authors comment on the levels of PSA (KLK3) in cell line-derived EVs (from different cell lines) in comparison to urine EVs?

Reviewer #2 (Remarks to the Author):

In their search for urinary biomarkers for prostate cancer, Khoo et al. implement differential ultracentrifugation to study the urinary proteome aiming to screen and compare the secreted proteome and extracellular vesicle proteome across men diagnosed or not with prostate cancer. The extracellular vesicle proteome is further assessed in two subgroups: extracellular vesicles obtained with a 20,000g centrifugation step (P20-EVs) versus extracellular vesicles obtained with a 150,000 g step (P150-EVs). The authors compare the secreted proteome and extracellular vesicle proteome before and after digital rectal examination, concluding that digital rectal examination overall enriches urine with prostate proteins. Next, the authors demonstrate that biophysical characteristics of P20-EVs and P150-EVs or the protein counts of the secreted proteome do not differ between urine samples from men diagnosed or not with prostate cancer, while the proteome does with urinary EVs reflecting the prostate tissue proteome. The authors further aim to identify the biomarker potential of the secreted proteome and extracellular vesicle proteome, concluding that both hold biomarker potential with thousands of secreted and extracellular vesicle proteins differing between men diagnosed or not with prostate cancer. Finally, the authors switch towards identifying some hits that can be exploited to develop affinity-based capture assays to rapidly assess a selection of potential urinary biomarkers. This study is well-written and well-structured. However, the manuscript overall lacks novelty, does not provide substantial biological insights nor brings substantial advances towards biomarker development compared to the current state-of-the-art in the field which is not properly discussed.

Main comments

-The authors describe throughout the manuscript the use of 'a simple protocol'. However, differential ultracentrifugation is used throughout the manuscript, which is overall not simple and suffers from issues related to reproducibility but also purity. Reproducibility is important to consider, especially when evaluating the proteome of an extended amount of samples for biomarker development. It is unclear how this has been controlled for throughout the study when analyzing 190 samples and how this could affect the results described in the manuscript.

-Although differential ultracentrifugation is used, the authors claim that "many urine EV proteins were identified as EV cargo in previous studies", that however seem to use more stringent methods compared to differential ultracentrifugation. This is rather striking, especially considering that the authors indicate (line 153) "Only 116 prostate-derived proteins were identified in unfractionated urine, whereas 2,439 were identified only in one or both of EV fractions and 4,968 in both EV and non-EV urine." This requires further analysis. In addition, these results should be further assessed considering the presence of potential protein corona formation.

-The difference between P20-EVs and P-150 EVs in terms of biophysical parameters but also protein content should be further explored and can contribute novel insights in both subgroups of extracellular vesicles in urine. Overall, this part of the manuscript remains preliminary and further analysis and validation is required.

-The authors do a proteome assessment of the secreted and extracellular vesicle urinary proteome, but in the end they do not really use this information to extract valuable biomarker information, which could have been the main strength of this study. First a focus is made on the 13 best available EV markers from literature. It is unclear why these markers are referred to as the best available EV markers and why exactly these were selected. More in-depth comparative analysis of the secreted proteome and P20-EVs and P150-EVs proteome in search of hits between men diagnosed with prostate cancer or not would have been more valuable. Some groups missing include other tumors in organs from the urogenital tract such as bladder.

-Overall, the last two sections of the manuscript with a focus on biomarker selection and identification of hits for affinity-based analysis of secreted proteome, P20-EVs and P150-EVs remains preliminary and requires further analysis.

Reviewer #3 (Remarks to the Author):

The study provides comprehensive proteome profiles from multiple different urine fractions including soluble, small, and large urinary EVs from patients and EVs from prostate cancer cells. They assessed potential biases in EV biophysical characteristics and tissue specific protein expressions by DRE using pre- and post- urine samples. This showed no significant biases and increases of prostate specific proteins as physically perturbed prostate. uEV from patient urines display better correlation with prostate tissue context compared to soluble fractions and cEV proteome. In addition, proteome from the large uEV (uEV-P20) exhibits better correlation with prostate and disease specific contexts due to a larger protein cargo. Selected marker candidates were also detected with independent affinity-based EV enrichment method to assess a potential of the markers for translation utility.

The study highlights that patient-derived EV should be prioritized in translation studies. However, soluble fractions should have possible indications including blood driven immune responses, secreted proteins from other organs. These could be an alternative indicator of the cancer related immune responses and disease complications.

The study used proteome information perturbed by DRE due to the simplicity to enrich prostate proteins. However, this is not application in RP condition. Thus, it is more interesting to see whether the selected marker candidates exhibit similar level of significance without DRE.

The study confirms utility of urinary EVs in translational and biomarker studies through a comprehensive proteome analysis by analyzing multiple different fractions. This actually increased the comprehension of the proteome from different fractions and exhibits proteome-wide heterogeneity in each fraction.

The proteome from patients were not compared with benign prostatic conditions such as BPH. For example, Elevated levels of certain urinary proteins like PCA3 are not specific to prostate cancer and can also be seen in other benign prostate conditions. The prostate specific proteins are likely detected in benign conditions, as well. And lack of validation with independent patient samples. There is discrepancy in EV selectivity and protein detection sensitivity between centrifugation-based and affinity-based methods. Factors like hydration status, time of day, and differences in collection and storage can affect urinary protein levels and make results less consistent. These points should be at least discussed or addressed by the experiments at some part.

RESPONSE TO REVIEWERS' COMMENTS

SUMMARY OF MAJOR CHANGES

As part of this revision, we have made to following major changes to our manuscript:

1. Additional validation of proposed biomarkers in independent cohorts. We added mass spectrometry-based proteomics from 299 independent patients (499 new mass spectrometry runs – *for details see below*).
2. Further analysis on EV subgroups: We thank reviewer 1 and 2 for their suggestion and have added the results from this analysis to Supplementary Table 4.
3. We have added text to discuss the caveats of differential ultracentrifugation for EV isolation as opposed to affinity-based approaches to the Discussion.
4. Source code, source data and new data generated for this revision have been made available at the respective data sources.

Passwords for access to data repositories and the web portal are as follows:

- Raw proteomics data files in MassIVE database (<ftp://massive.ucsd.edu>): #MSV000092061 (Password: urineproteome23)
- Custom web portal for data exploration: <http://kislingerlab.uhnres.utoronto.ca/ev/home/>. (Password: urineproteome23)

REVIEWER COMMENTS

Reviewer #1 (Remarks to the Author):

In this well-written manuscript, the authors generated comprehensive proteomic profiles of urine (urine soluble proteins (uSP) and urinary extracellular vesicle (uEV) proteins) from a mixed cohort of 190 men, including a subset with prostate cancer. They first show in matched pre- and post-digital rectal examination (DRE) urine from 10 men with low grade prostate cancer that DRE increases abundance of prostate-derived proteins in all urine fractions (uSP, uEV-P20, and uEV-P150) without influencing EV biophysical characteristics. They then analyze proteomes of post-DRE urine from 190 men, of whom 64 had no prostate cancer and 126 had untreated prostate cancer (54 with GG1, 20 GG2, 26 GG3, 11 GG4, 15 GG5). They characterize the protein cargo in different urine fractions and find that urine EVs are more reflective of the prostate tissue proteome, in particular uEV-P20. When compared with prostate cancer cell line-derived EVs, urine EVs have similar biophysical characteristics, but have protein composition more reflective of prostate tissues. Next they study a longitudinal cohort of 5 prostate cancer patients and identify EV proteins that are stable over time. They then compare urine proteomes of patients with and without prostate cancer, and then use machine learning to create classifiers to distinguish cancer from non-cancer, and to distinguish GG1 (low grade) from GG>1 (higher grade) cancer. Finally, they performed bioinformatic analyses to identify optimal protein markers that potentially could be used for affinity-based isolation of specific actionable EV subpopulations.

Overall this is a comprehensive analysis of urine EV proteomics in a large cohort of men with or without prostate cancer, with potential clinical implications for the development of minimally invasive diagnostic and active surveillance tools. The proteomics of urine EVs and their potential clinical utility has been studied extensively in the past with multiple publications in the literature, but this work is significant for the size of cohort and the depth of the proteomic analysis. The

development of urine proteomic classifier models is especially interesting, although a separate validation cohort is missing. Additional comments are as follows:

We thank the reviewer for recognizing the value of our work.

1. Line 127-128 and Ext. Data Fig 2C: The text states that “uEV biophysics ... were largely independent of disease status...”, but Ext. Data Fig. 2C shows that GG3 has more particles than the other grade groups (and significantly more than GG4-5). This might be a spurious finding due to low numbers of patients in each group, but the authors should address this in the text.

We thank the reviewer for the suggestion. We have clarified the text to discuss this point. The text on Lines 133-137 on page 7 now reads:

“uEV biophysics and urine protein counts were largely independent of disease status, age or serum PSA levels (**Supplementary Fig. 2a-g**). We observed an increasing trend in uEV particle count and cISUP GG 1-3 followed by a decrease in particle count in cISUP GG 4-5, although this finding requires further validation in larger cohorts due to the low numbers of patients in each group (**Supplementary Fig. 2c**).”

2. Line 132 and Ext. Data Fig. 3A: Although the majority of urine EV proteins identified in this study overlapped with those identified in previous studies, it appears some proteins were unique to this study. Were any of them uniquely found only in certain patient subsets such as GG4 and GG5 cancers? Were these newly identified urine EV proteins of any biological interest

We thank the reviewer for this suggestion and agree that further investigation of proteins unique to this study could be interesting. When doing this analysis we found an error in our code where the incorrect data was used to make the original upset plot in **Supplementary Fig. 3** (see **Rebuttal figure 1A**). We apologize for the oversight and have replaced the figure with the correct data (see **Rebuttal figure 1B**). The source data and source code for reproducing this analysis and figure are provided with this paper.

Rebuttal figure 1: Original (left) and new (right) Supplementary Fig. 3c (right) with correct dataset.

We further investigated whether there were proteins uniquely detected in certain cISUP Grade Groups or in non-cancer patients. We examined each subset of unique proteins from **Supplementary Fig. 3c**:

- uEV-P20 unique: Proteins unique to our uEV-P20 dataset and not detected in our uEV-P150 dataset, Exocarta, or Vesiclepedia databases (1,173 proteins)
- uEV-P150 unique: Proteins unique to our uEV-P150 dataset and not detected in our uEV-P20 dataset, Exocarta, or Vesiclepedia databases (286 proteins)
- uEV-P20/uEV-P150 unique: Proteins unique to uEV-P20 or uEV-P150 in this study but not Exocarta, or Vesiclepedia databases (881 proteins)

We found that majority of the proteins are shared across clinical groups (**Rebuttal figure 2**). We have made this data available in **Supplementary Table 4 Sheet 4d**.

Rebuttal figure 2: Proteins unique to this study in each cISUP Grade Group. Top: Supplementary Fig. 3C.

Of the proteins unique to our study, we further considered proteins present in >50% of each respective fraction and examined whether they are differentially abundant in prostate cancer (PCa) and non-cancer (NC) patients. We identified many proteins that were unique to our study and differentially abundant in PCa and NC patients, suggesting that the proteins unique to our study have potential biological interest (**Rebuttal Figure 3** and **Supplementary Table 4**). Altogether, we show that we were able to detect EV proteins unique to our study that were of potential biological interest.

Rebuttal figure 3: Differential abundance of proteins unique to this study in prostate cancer and non-cancer patients.

We have summarised these results in **Supplementary Table 4** Sheet 4d and 4e. For the question on whether there were proteins unique to each grade group, we added the following text to the **Results** section on **page 8, lines 145-146**:

“Of the 6,518 proteins detected in uEVs, 60% were identified as EV cargo in previous studies (**Supplementary Fig. 3c-d**)^{2,17,20–22}.”

For the question of whether proteins uniquely detected in our study were of biological interest, we added text to the **Results** section on page 11-12, lines 230-236:

“Of these proteins that were differentially abundant in prostate cancer and non cancer uEVs, 541 proteins were uniquely detected in our dataset compared to other datasets of post-DRE urine-derived uEVs from patients with prostatic disease^{17,33,34}(**Supplementary Table 4**). In addition, only 21 proteins were unique to a single disease subgroup (*i.e.* non-cancers or specific cISUP Grade Groups). These results suggest that differences in disease groups are reflected in the differential abundance of proteins in the uEV proteome.”

3. Fig. 5A-B: Although the longitudinal evaluation and protein variance analysis over time is interesting, this analysis is very limited given that only 5 patients are included.

We agree with the reviewer that the cohort is small for longitudinal analysis. However, longitudinal urines for the same patient is difficult to acquire due to the requirement for serial urine collections from the same patient with repeat needle biopsies to confirm no upgrading. In this study, we restricted our analysis to patients that all have cISUP GG 1 tumors **and** similar serum PSA levels **and** who did not upgrade over a time period of 1-2 years. Therefore, observed differences in the

proteome can be attributed to the stability of protein measurements over time and not differences due to tumor upgrading.

To the **Discussion** on page 16, lines 326-330, we added text that mentions that the small cohort size and the need for verification in larger, more clinically heterogeneous cohorts.

“Our analysis of temporal stability of the urine proteome was restricted to prostate cancer patients that all have cISUP GG 1 tumors, similar serum PSA levels, and who did not have clinical upgrading over a period of 1-2 years. Follow-up studies utilizing larger, clinically heterogeneous cohorts will be needed to verify if the temporally stable proteins identified in low-grade patients generalize to other patients with other tumor grades.”

4. Lines 212-215 and Figs. 5C-E: The finding that PSA (KLK3) protein abundance is consistently reduced in all fractions of urine from prostate cancer patients compared to men without cancer, despite being increased in serum and tumor, is puzzling. The authors should provide an explanation for this finding in their Discussion.

We thank the reviewer for their suggestion and have clarified the text to discuss this finding. This observation is consistent with what other groups have observed and we have updated the **Discussion** on **page 16-17, lines 332-337**. The text now reads:

“Consistent with data from other groups, prostate proteins such as PSA were elevated in the urine and urinary EVs of patients without cancer compared to those with prostate cancer, while serum PSA levels display the opposite trend^{33,34,43}. This is likely due to the disruption of prostate cellular architecture with increasing tumor grade, which results in elevated levels of PSA in the bloodstream and consequently decreased levels of PSA in urine^{33,44}.”

5. The urine proteomic classifiers (Figs. 5G-H) for distinguishing cancer vs. non-cancer and GG1 vs. GG>1 are promising, but they have been trained and evaluated in the same patient cohort using leave-one-out-cross validation. It will be necessary to validate these classifier models in separate independent patient cohorts.

We agree with the reviewer that additional validation of these classifier models in independent patient cohorts is indeed necessary. We have acquired three independent validation cohorts for the validation of the PCa vs NC models for all three urine fractions (Validation cohort 1: 30 patients) and validation of the GG1 vs GG>1 uSP classifier (Validation cohort 2: 199 patients and Validation cohort 3: 70 patients).

In this revision, Reviewer 2 suggested that we did not fully leverage our proteomic data that was collected on discrete urinary fractions to extract information that could better inform biomarker discovery. We agree with the reviewer that we could have used a more stringent selection process and have made changes to our biomarker discovery pipeline to include this information/suggestion. Due to the new figures generated, the original **Figure 5** has been split into new **Figures 5 and 6**.

Our modified biomarker discovery pipeline now includes several new inclusion criteria (**Figure 6a**) for selecting the most relevant proteins for machine learning. In addition to the existing inclusion criteria of protein detection in >50% of urines and longitudinally stable proteins, our new inclusion criteria include enrichment of a protein in a specific urinary fraction (from differential expression analysis in **Figure 2j**), as well as protein detection in more than half of prostate tissues

(suggesting that the protein is likely prostate-derived). These resulting proteins (uSP: 228, uEV-P20: 281, uEV-P150: 236) were then used as features for building classifiers by machine learning (new **Figure 6a**).

Figure 6a: New pipeline for biomarker discovery.

For validating prostate cancer/non-cancer signatures, we acquired urines from a cohort of 30 patients, of which 16 had prostate cancer and 14 had benign prostatic hyperplasia (non-cancer). We report validation AUCs from 0.71-0.81 and concordant effect sizes (\log_2 fold change) between discovery and validation datasets for all urinary fractions (**Figures 6b-d**).

Figures 6b-d: Performance of PCa/NC classifiers in discovery (B) and validation (D) datasets with effect sizes (C) of proteins in the classifiers.

We recognize that the size of validation cohort is small and we would like to note that BPH post-DRE urines are not typically bio-banked in prostate cancer clinics. In addition, for uEV proteomics analysis, we require 14 mL of urine per patient for uEV isolation, which is the entirety of the average volume of first-catch first-flow post-DRE urine (~15-20 mL). These large volumes of post-DRE urine are rarely available and only accessible after years of urine collection.

For verifying cISUP GG>1 vs GG=1 classifiers, we assembled independent validation cohorts consisting of 199 patients (Validation 2) and 70 patients (Validation 3) with prostate cancer (**Supplementary Table 1**). These samples were collected, processed and data acquired at two different institutes (Toronto or Norfolk, Virginia). We report concordant effect sizes for 5 proteins in 2/3 datasets and 3 proteins in 3/3 datasets (**Supplementary Fig. 6b**). The remaining proteins had small effect sizes in the discovery cohort. Our power analysis revealed that we were underpowered to validate the remainder of the proteins due to the limited size of the cohort (**Rebuttal Table 2**). Due to the large volumes of urine and cohort size required for validation of the GG1 vs GG>1 classifier for uEVs we were not able to validate that classifier at this point in time. Further validation of these candidate proteins in larger cohorts >300 samples will be required, should these larger cohorts be accessible in the future. We further discuss this point in our **Discussion** page 17-18 where we highlight the need for validation in larger, diverse cohorts and the need for validation using targeted mass spectrometry assays.

Supplementary Fig. 6b: Effect sizes of proteins in uSP cISUP>1/cISUP=1 protein signature in discovery and validation cohorts.

Classifier	gene	sd (log2)	fc (log2)	Standardized fc (log2)	Sample size needed (Tolerance = 0.05)	Sample size needed (Tolerance = 0.10)
uSP	CAT	2.02	2.08	1.03	88	65
uSP	CADM1	1.00	-0.20	-0.20	NA	NA
uSP	NUTF2	0.81	-0.26	-0.32	>300	>300
uSP	GLG1	1.45	-0.53	-0.37	>300	>300
uSP	SERPINA1	1.47	-0.67	-0.46	>300	>300
uSP	CTSB	0.97	-0.51	-0.53	273	188
uSP	ITFG1	1.25	-0.74	-0.60	220	154
uSP	CST3	0.97	-0.70	-0.72	159	113
uSP	SPOCK1	2.50	-1.94	-0.77	141	101
uSP	MIEN1	0.93	-0.91	-0.98	95	69
uSP	PEBP4	1.36	-1.35	-0.99	94	69
uSP	KLK3	2.23	-3.08	-1.38	55	42
uEVP20	HBG2	1.92	2.66	1.39	58	44
uEVP20	OCIAD2	1.72	2.01	1.17	75	57
uEVP20	ITGAM	2.97	1.90	0.64	208	148
uEVP20	CARHSP1	1.13	0.36	0.32	>300	>300
uEVP20	RPLP1	1.35	0.19	0.14	>300	>300
uEVP20	KRT78	3.84	0.44	0.11	>300	>300
uEVP20	MB	1.65	-0.64	-0.39	>300	>300
uEVP20	RPS20	1.09	-1.06	-0.97	104	77

uEVP150	CKAP5	3.08	0.50	0.16	NA	NA
uEVP150	TACC2	3.60	0.18	0.05	NA	NA
uEVP150	GSTM4	4.09	-0.29	-0.07	NA	NA

Rebuttal Table 2: Estimated cohort sizes needed for robust determination of observed effect sizes in cISUP Grade Group signatures. Number of genes for each fraction: 3238 (uSP), 5686 (uEV-P20) and 4067 (uEV-P150). Assumed population prevalence of 0.57 in the largest group. NA: Standardized fold change is too small (<0.2) to estimate a sample size. Standardized fold change is calculated as the difference between groups divided by the within-class standard deviation, on the log2 scale. Tolerance: "The sample size n is chosen to ensure that the expected (average) accuracy of the classifier is within the tolerance of the best possible classifier" (from NCI Biometric Research Program guidelines for developing classifiers).

We added the following text to the **Results** on **page 12-13, lines 246-255 and 260-263**:

"To create biomarkers of prostatic disease, we focused on proteins that were frequently detected (in >50% of samples), enriched in a urinary fraction, prostate-derived, and longitudinally stable (ICC > 0.4) (**Figure 6a**) This filtering strategy retained 226 uSP proteins, 280 uEV-P20 proteins and 235 uEV-P150 proteins. We used statistical machine learning to create and validate classifiers independently for each urine fraction. First, we created classifiers that distinguish cancer from non-cancer based solely on urine proteins; these had AUCs ranging from 0.92-1, significantly outperforming serum PSA (**Figure 6b**). In an independent, prospective validation cohort of 30 patients with proteomic data from all three urinary fractions (**Supplementary Table 1**), proteins had concordant effect sizes and validation AUCs ranging from 0.71-0.81 (**Figures 6c-d**)"

"In two independent validation cohorts of 199 and 75 patients (**Supplementary Table 1**), effect sizes of uSP proteins between the discovery and validation cohorts were concordant (**Supplementary Fig. 6b**)."

We added the following text to the **Discussion** on **page 17-18, lines 356-366**:

"Since our unique dataset represents one of the largest urine proteomics studies for prostate cancer, we were able to leverage urinary fraction-specificity, frequency of detection in prostate tissues and longitudinal stability to prioritize proteins as new biomarker candidates for prostatic diseases. The proteomics dataset generated in this study comprises one of the larger-sized discovery cohorts for urinary proteomics studies totaling >100 patients for prostate cancer biomarker discovery in urine, even for urinary EVs, and is well powered to detect log₂ fold changes in protein abundances of 1. However, while many of the proteomic differences between groups (e.g. cISUP Grade Groups) are concordant in independent cohorts, these differences are small. Further validation of the urinary signatures using robust quantitation by targeted proteomics assays with stable isotope labeled standards^{2,50} in larger, racially diverse validation cohorts is needed."

6. It is interesting that among the various urine proteomic classifiers, the urine soluble protein (uSP) based models actually perform very well with a high AUC of 0.94 for cancer vs. non-cancer, and performs even better than the urine EV protein-based classifiers for GG1 vs. GG>1 (AUC 0.86 for uSP, compared to 0.79 for uEV-P20 and 0.67 for uEV-P150). This could be used to argue that EVs don't need to be isolated from urine at all, and that a simple urine test based just on just soluble proteins could be more useful as a biomarker as opposed to urine EVs.

The reviewer raises an interesting point as to which urinary fraction would be most useful as a biomarker. We agree with the reviewer that a test using soluble proteins would be simpler than one using urine EVs. However, when we performed bootstrapping to assess whether differences between AUCs were statistically significant, we found that they were not, and therefore we are unable to comment on which urinary fraction is the best.

In **Supplementary Table 4** Sheet 4i, we have added a table summarizing the performance of the models and how they compare to each other.

Model comparisons				
PCa/NC signatures (Discovery)				
	AUC1	AUC2	Difference in AUC	P-value ²
uSP vs uEV-P20	1	0.928	0.072	0.041
uSP vs uEV-P150	1	0.917	0.083	0.129
uEV-P20 vs uEV-P150	0.928	0.917	0.011	0.865
PCa/NC signatures (Validation)				
	AUC1	AUC2	Difference in AUC	P-value ²
uSP vs uEV-P20	0.728	0.808	0.08	0.483
uSP vs uEV-P150	0.728	0.71	0.018	0.891
uEV-P20 vs uEV-P150	0.808	0.71	0.098	0.243
cISUP signatures (Discovery)				
	AUC1	AUC2	Difference in AUC	P-value ²
uSP vs uEV-P20	0.788	0.787	0.001	0.992
uSP vs uEV-P150	0.788	0.725	0.063	0.504
uEV-P20 vs uEV-P150	0.787	0.725	0.062	0.519

Supplementary Table 4 Sheet 4i: Summary of model comparisons for each clinical comparison.

7. Fig. 4: Can the authors comment on the levels of PSA (KLK3) in cell line-derived EVs (from different cell lines) in comparison to urine EVs?

We thank the author for the suggestion and have added this information to **Supplementary Table 4 Sheet 4f**.

Most of the commonly available cell lines for prostate cancer vary in androgen sensitivity and thus many cell lines do not express PSA at the transcript level and/or protein level. PSA protein levels in our whole cell lysate proteomics analysis mirror the expected phenotypes of these cell lines and are summarised in **Supplementary Table Sheet 4f**. With the exception of 22Rv1, where PSA can be detected in whole cell lysates (WCL) but not cell line EVs (cEV), the cell line EVs from the other cell lines mirror the PSA expression levels of the whole cell lysates.

Cell line	Characteristics (from ATCC)	PSA in WCL (our data)	PSA in cEV-P20	PSA in cEV-P150
22Rv1	Expresses PSA	Yes	No	No
DU145	Does not express PSA	No	No	No
LNCaP	Expresses PSA	Yes	Yes	Yes
PC3	Does not express PSA	No	No	No
RWPE1	Does not express PSA without androgen induction	No	No	No

Supplementary Table 4F: Prostate specific antigen expression in cell line-derived extracellular vesicles.

We have added the following text to the **Results** section on **page 10, lines 197-201**:

“PSA protein expression in whole-cell lysates was consistent with the androgen sensitivity of the cell lines, with androgen insensitive cell lines DU145, PC3 and RWPE1 not expressing PSA and androgen sensitive cell lines 22Rv1 and LNCaP expressing PSA³⁴. In cEVs, only EVs isolated from LNCaP had detectable PSA (**Supplementary Table 4**).”

Reviewer #2 (Remarks to the Author):

In their search for urinary biomarkers for prostate cancer, Khoo et al. implement differential ultracentrifugation to study the urinary proteome aiming to screen and compare the secreted proteome and extracellular vesicle proteome across men diagnosed or not with prostate cancer. The extracellular vesicle proteome is further assessed in two subgroups: extracellular vesicles obtained with a 20,000g centrifugation step (P20-EVs) versus extracellular vesicles obtained with a 150,000 g step (P150-EVs). The authors compare the secreted proteome and extracellular vesicle proteome before and after digital rectal examination, concluding that digital rectal examination overall enriches urine with prostate proteins. Next, the authors demonstrate that biophysical characteristics of P20-EVs and P150-EVs or the protein counts of the secreted proteome do not differ between urine samples from men diagnosed or not with prostate cancer, while the proteome does with urinary EVs reflecting the prostate tissue proteome. The authors further aim to identify the biomarker potential of the secreted proteome and extracellular vesicle proteome, concluding that both hold biomarker potential with thousands of secreted and extracellular vesicle proteins differing between men diagnosed or not with prostate cancer. Finally, the authors switch towards identifying some hits that can be exploited to develop affinity-based capture assays to rapidly assess a selection of potential urinary biomarkers. This study is well-written and well-structured.

However, the manuscript overall lacks novelty, does not provide substantial biological insights nor brings substantial advances towards biomarker development compared to the current state-of-the-art in the field which is not properly discussed.

We thank the reviewer for their suggestions and have made substantial changes to the biomarker development portion of the manuscript to better make use of the urinary EV data.

However, we disagree with the reviewer that the study does not bring substantial advances towards biomarker development. First, of all the existing discovery proteomics studies that utilize post-DRE urines or urinary EVs for proteomic characterization of localized prostate cancer (*i.e.* non-metastatic), our study has the largest cohort with the highest depth of proteome coverage (**Rebuttal Table 1**). Our cohort has also been carefully selected for several clinical variables for each comparison. These include:

- In **Figure 1**, for pre-DRE vs post-DRE comparisons, matched urines were collected for patients with cISUP Grade Group 1 disease. Therefore, differences observed in the urinary proteome can be attributed to the DRE and not serum PSA, which is not controlled for in other studies. We show that post-DRE urine increases the number of prostate proteins that can be detected in urine.
- In **Figure 5**, we acquired data from longitudinal uEV and uSP proteomes over the period of 12-24 months. To our knowledge, this is the only longitudinal analysis of post-DRE urine and EV proteins for prostate cancer patients. In addition, to rule out variability in the proteome that could be attributed to disease state, we only

included patients with cISUP Grade Group 1 cancers on active surveillance with no upgrading (assessed by needle biopsy) or changes in serum PSA levels over time. While there are other assessments of urine and urinary EV stability (Nagaraj and Mann 2011 *J Proteome Res* and Erozcenci et al. 2021 *Sci Rep*), none have assessed the stability of post-DRE urines, which can differ from non-DRE urines due to prostatic massage. From this analysis, we were able to identify proteins in post-DRE urine that had high variance in protein abundance over time for exclusion as biomarkers.

- In **Figure 5**, for comparisons of post-DRE urine proteomes in patients with and without prostate cancer, prostate cancer patients were selected to exclude those with very high serum PSA (> 100 ng/mL). The cohort was also biased towards the most-clinically relevant patients with low- (cISUP Grade Group 1) or low/intermediate-grade cancers (cISUP Grade Group 2) and who had serum PSA levels in the “grey” range of 4-10 ng/mL. In this serum PSA range, the PSA test has poor specificity for prostate cancer and is therefore a clinical group where a fluid biomarker would have the most impact. In this analysis, we identified hundreds of differentially abundant proteins per fraction that had potential as fluid biomarkers for PCa/NC.
- In **Figures 2 and 5-7**, we used the differential abundance of proteins as well as the frequency of detection to only consider proteins that are reproducibly detected by mass spectrometry. This is an important consideration as for many mass spectrometry-based protein biomarkers, further validation with targeted mass spectrometry assays is needed. In this cohort, this is a strict requirement as proteins must have a measurable quantity in >75 samples. This requirement and large sample size allows us to determine the variability of the protein measurements as well as frequency of detection in each group, which will enable selection of the best candidates for future validation with targeted mass spectrometry assays.

Post-DRE urine study	Post-DRE Urine (patients)	Non-cancer (BPH or BiopsyNegative) patients	# Proteins
Urine soluble proteins (uSP)			
Our study	379	54	3,238
Wang et al 2022 Prostate (PMID: 36479985)	292	0	585 (Glycoprotein only)
Lih et al 2020 Sci Rep (PMID: 36050450)	142	0	4 (targeted proteomics only)
Principe et al 2012 J Proteome Res (PMID: 22339264)	10	5	1,022
Shrivastava et al 2020 Indian J Urol (PMID: 31983827)	77	45	2 (targeted proteomics only)
Extracellular vesicles (uEV)			

Our study	183	39	5,686 (uEV-P20: dUC @ 20,000 g) 4,067 (uEV-P150: dUC @ 150,000 g)
Dhondt et al 2020 (PMID: 32284825)	24	12	2,333 (all fractions: EV, THP and protein) 623 (EV only: dgC @ 100,000 g)
Fujita et al 2017 (PMID: 28211531)	18	6	4,710 (by dUC @ 17,000 g then 100,000 g)

Rebuttal Table 1: Cohort sizes in other studies utilizing post-DRE urine from patients with prostatic diseases. dUC: Differential ultracentrifugation, dgC: Differential gradient centrifugation.

Main comments

-The authors describe throughout the manuscript the use of ‘a simple protocol’. However, differential ultracentrifugation is used throughout the manuscript, which is overall not simple and suffers from issues related to reproducibility but also purity. Reproducibility is important to consider, especially when evaluating the proteome of an extended amount of samples for biomarker development. It is unclear how this has been controlled for throughout the study when analyzing 190 samples and how this could affect the results described in the manuscript.

We thank the reviewer for their suggestion. We agree with the reviewer that EV isolation by differential ultracentrifugation is not a simple protocol. However, we would like to clarify that our text refers to the digital rectal exam as a simple protocol for enriching for prostate proteins in urine, and that our reference to “simple protocol” does not refer to EV isolation which is indeed complex. The text currently reads (**page 6, lines 103-106**):

“DREs are routine, minimally-invasive physical manipulations performed millions of times annually by oncologists and primary care physicians, and thus might provide a simple approach to enrich prostate-derived proteins in urine.”

We agree with the reviewer that the reproducibility of uEV isolation by differential ultracentrifugation is definitely important to consider prior to biomarker development. To isolate urinary EVs in this study, we used a published protocol by Correll et al⁵ who assessed the reproducibility of EV isolation from post-DRE urine and found it to be highly reproducible Spearman’s $\rho > 0.99$. We also independently repeated their protocol using independent post-DRE urine samples - 3 samples from 3 patients with cISUP Grade Group 1 prostate cancer, 1 pooled sample from 10 non-cancer patients with benign prostatic hyperplasia, and 1 pooled sample from 10 non-cancer patients who had elevated serum PSA levels (serum PSA > 4 ng/mL) but no cancer detected on transrectal ultrasound-guided needle biopsy). In our cohort, the median Spearman’s ρ for processing replicates for uEV-P20 and uEV-P150 was 0.941 (range 0.890 – 0.975) and 0.948 (range 0.925 – 0.972), respectively. Proteomes of processing replicates were more similar to each other than between biological groups (e.g. Prostate cancer *versus* non-cancer), suggesting that our uEV isolation protocol is highly reproducible and able to uncover differences between clinical groups.

We have added this analysis to **Supplementary Fig. 3**. We have also updated the main text on **page 7** lines 143-146 to read:

“uEV isolation was also highly reproducible, with larger proteomic differences observed between patients and clinical groups compared to within samples (**Supplementary Fig. 3a-b**).”

Supplementary Fig. 3a and 3b: Reproducibility of uEV isolation from post-DRE urine.

-Although differential ultracentrifugation is used, the authors claim that “many urine EV proteins were identified as EV cargo in previous studies”, that however seem to use more stringent methods compared to differential ultracentrifugation. This is rather striking, especially considering that the authors indicate (line 153) “Only 116 prostate-derived proteins were identified in unfractionated urine, whereas 2,439 were identified only in one or both of EV fractions and 4,968 in both EV and non-EV urine.” This requires further analysis.

We agree with the reviewer that differential ultracentrifugation (dUC) is indeed a less stringent method for isolating urinary EVs compared to other methods such as density gradient centrifugation (dGC) which was used in a previous study by Dhondt *et al*. However, we would like to clarify that we had compared the dUC-uEV proteomes from our study to proteins in publicly available EV databases (**Supplementary Fig. 3c**) as well as dGC-uEVs from Dhondt *et al* 2020 and two other uEV studies (**Supplementary Fig. 3d**) to verify that proteins detected in the uEV fractions in our study are expected uEV proteins. Since we recognize that dUC is a crude isolation protocol and is not as specific for uEVs compared to other non-EV urine-soluble proteins, we followed up with additional comparison of uEV proteomes to urine soluble (uSP) proteomes using matched samples to identify uEV-enriched proteins (**Figure 2h-j**).

In addition, these results should be further assessed considering the presence of potential protein corona formation.

In this study, we washed each uEV fraction with 100 mM Na₂CO₃ (pH 11.0) to reduce the protein constituents of the protein corona (Correll *et al* 2020 *J Extracellular Vesicles*)⁵. This method has previously been shown to deplete high abundance urine proteins (e.g. uromodulin, albumin), non-EV contaminants, and ECM proteins from uEVs compared to without washing. In that study, the

authors also showed that by depleting high abundance proteins in the protein corona, more proteins can be detected by mass spectrometry. Since the assessment of protein corona components has already been discussed in depth in a published manuscript⁵, we did not include this assessment in this study.

-The difference between P20-EVs and P-150 EVs in terms of biophysical parameters but also protein content should be further explored and can contribute novel insights in both subgroups of extracellular vesicles in urine. Overall, this part of the manuscript remains preliminary and further analysis and validation is required.

We have highlighted throughout our manuscript how the protein content of uEV-P20 and uEV-P150 differ.

- In **Figure 2h-j** we showed how proteins in the different uEV fractions differed in frequency of detection as well as in protein abundance and followed up with an analysis of the predicted subcellular localization of these fraction-enriched proteins in **Figure 2k**. These data highlight how uEV-P20 and uEV-P150 cargo differ in subcellular localization.
- In **Figure 3** we then show how the proteome of uEVs compare to tissue in proteins detected (**Figure 3a**) and in protein abundance (**Figure 3b**). This analysis is then followed up with a pathway analysis to highlight the subcellular compartments from which these sample-type enriched proteins originate compared to the tissue proteome. These data indicate that uEV-P20 and uEV-P150 recapitulate different components of the prostate tissue proteome.
- In **Figure 4** we do several comparisons of cell line-derived EV-P20 to EV-P150, as well as how they compare to urine-derived EVs, including pathway analysis. These data suggest that some of the EV-P20 and EV-P150 proteome differences are conserved in urine and cell lines.

We've also made available the complete list of differentially abundant proteins for each of these comparisons in **Supplementary Table Sheet 4c**.

-The authors do a proteome assessment of the secreted and extracellular vesicle urinary proteome, but in the end they do not really use this information to extract valuable biomarker information, which could have been the main strength of this study.

We thank the reviewer for this suggestion and agree that not using this information to extract valuable biomarker information is a missed opportunity. We have made changes to our biomarker discovery pipeline to fully leverage this information. Due to the new figures generated, the original **Figure 5** has been split into new **Figures 5 and 6**.

Our modified biomarker discovery pipeline now includes several new inclusion criteria (**Figure 6a**) for selecting the most relevant proteins for machine learning. In addition to the existing inclusion criteria of protein detection in >50% of urines and longitudinally stable proteins, our new inclusion criteria include enrichment of a protein in a specific urinary fraction (from differential expression analysis in **Figure 2j**), as well as protein detection in more than half of prostate tissues (suggesting that the protein is likely prostate-derived). These resulting proteins (uSP: 226, uEV-P20: 280, uEV-P150: 235) were then used as features for building classifiers by machine learning (new **Figure 6a**).

Figure 6a: New pipeline for biomarker discovery.

First a focus is made on the 13 best available EV markers from literature. It is unclear why these markers are referred to as the best available EV markers and why exactly these were selected.

We apologise for the confusion in the text and have clarified the text to describe these markers. The text on page 14, lines 284-285 now reads:

“We evaluated the performance of 13 EV markers that are ubiquitous to multiple human cancer tissues and fluids but not yet evaluated as prostate urinary EVs⁶.”

More in-depth comparative analysis of the secreted proteome and P20-EVs and P150-EVs proteome in search of hits between men diagnosed with prostate cancer or not would have been more valuable.

We thank the reviewer for their helpful suggestion. We have used PCa/NC differential abundance in each sample type to rank proteins for gene set enrichment analysis against the Hallmark gene sets in the human MSigDB collection⁷. The new dotmap (**Supplementary Fig. 5c**) shows GSEA Hallmark terms that were enriched in each sample type (FDR-adjusted P-value < 0.25).

Supplementary Fig. 5c: Dotmap of significant GSEA Hallmark terms in PCa/NC comparisons for all fractions.

We added text to the Results pages. 11-12 summarising this new result:

“Next, we sought to evaluate the biomarker potential of urine-soluble and uEV proteins for predicting prostatic disease. Men with benign prostatic conditions (non-cancer [NC]) included individuals with elevated serum PSA levels and benign prostatic hyperplasia as well as patients with no diagnosed prostate cancer on transrectal ultrasound-guided 12-core biopsy (**Supplementary Table 1**). Patients with and without prostate cancer had similar serum PSA abundances ($P = 0.34$). Despite this similarity, thousands of soluble (**Figure 5c**) and EV proteins (**Figures 5d-e**) differed between patients with and without cancer. The specific differentially abundant proteins varied between urinary fractions (**Supplementary Fig. 5b**). Of these proteins that were differentially abundant in prostate cancer and non-cancer uEVs, 541 proteins were uniquely detected in our dataset compared to other datasets of post-DRE urine-derived uEVs from patients with prostatic disease^{17,33,34} (**Supplementary Table 4**). In addition, only 21 proteins were unique to a single disease subgroup (*i.e.* non-cancers or specific cISUP Grade Groups). These results suggest that differences in disease groups are reflected in the differential abundance of proteins in the uEV proteome.”

Some groups missing include other tumors in organs from the urogenital tract such as bladder.

We thank the reviewer for this suggestion and agree that it is an interesting question to determine the tissue of origin for urinary proteins. However, we did not include urines from other tumors in the urogenital tract as post-DRE urine is not collected for patients without prostatic disease.

Instead, to determine if post-DRE urine is enriched in proteins of prostate origin compared to normal bladder or kidney tissue, in **Figure 3c-d** we use prostate, bladder and kidney tissue from patients without cancer (GTEx) or the normal adjacent tissue from cancer patients (TCGA) to identify tissue-elevated genes for an enrichment analysis. Here, we show that prostate proteins, and not kidney or bladder proteins, are enriched in uEV-P20 and uEV-P150 isolated from post-DRE urine.

In our new biomarker selection workflow (**Figure 6a**), we added an additional requirement that the protein in urine or uEVs must be detected in >50% of prostate cancer tissues (*i.e.* >75 tissues), which adds further confidence that the protein is prostate-derived and not derived from other tissues in the urogenital tract.

-Overall, the last two sections of the manuscript with a focus on biomarker selection and identification of hits for affinity-based analysis of secreted proteome, P20-EVs and P150-EVs remains preliminary and requires further analysis.

We agree with the reviewer that further validation of our identified biomarkers is needed. As such, we have performed additional validation of our proposed PCa/NC and cISUP biomarker signatures in independent validation cohorts from an additional 299 patients (499 mass spectrometry runs), bringing the totals of newly generated data for this study to 1,400 mass spectrometry runs from 539 patients.

For validating prostate cancer/non-cancer signatures, we acquired urines from a cohort of 30 patients, of which 16 had prostate cancer and 14 had benign prostatic hyperplasia (non-cancer). We performed proteomics on all three urinary fractions and report validation AUCs from 0.71-0.81 and concordant effect sizes (\log_2 fold change) between discovery and validation datasets for all urinary fractions (**Figures 6b-d**).

Figures 6b-d: Performance of PCa/NC classifiers in discovery (b) and validation (d) datasets with effect sizes (c) of proteins in the classifiers.

We recognize that the size of validation cohort for PCa/NC is small and we would like to note that BPH post-DRE urines are not typically bio-banked in prostate cancer clinics. In addition, for uEV proteomics analysis, we require 14 mL of urine per patient for uEV isolation, which is the entirety of the average volume of first-catch first-flow post-DRE urine (~15-20 mL). These large volumes of post-DRE urine are rarely available and only accessible after years of urine collection.

For verifying cISUP GG>1 vs GG=1 uSP classifiers, we assembled independent validation cohorts consisting of 199 patients (Validation 2) and 70 patients (Validation 3) with prostate cancer (**Supplementary Table 1**). These samples were collected, processed and data acquired at two different institutes (Toronto or Norfolk, Virginia). We report concordant effect sizes for 5 proteins in 2/3 datasets and 3 proteins in 3/3 datasets (**Supplementary Fig. 6b**). The remaining proteins had small effect sizes in the discovery cohort. Our power analysis (according to the NIH's National Cancer Institute's Biometric Research Program's guidelines) revealed that we were underpowered to validate the remainder of the proteins due to the limited size of the cohort (**Rebuttal Table 2**). Further validation of these candidate proteins in larger cohorts >300 samples will be required, should these larger cohorts be accessible in the future. We further discuss this point in our **Discussion** page 17-18 where we highlight the need for validation in larger, diverse cohorts and the need for validation using targeted mass spectrometry assays. Due to the difficulty in obtaining additional post-DRE urines at sufficient volumes (i.e. 14 mL) for EV isolation as mentioned above and the need for cohort sizes >300 patients (**Rebuttal Table 2**) to detect the

\log_2 fold changes observed for cISUP Grade Group >1 vs cISUP Grade Group = 1, we are unable to validate the cISUP Grade markers at this time.

Supplementary Fig. 6b: Effect sizes of proteins in uSP cISUP>1/cISUP=1 protein signature in discovery and validation cohorts.

Classifier	gene	sd (log2)	fc (log2)	Standardized fc (log2)	Sample size needed (Tolerance = 0.05)	Sample size needed (Tolerance = 0.10)
uSP	CAT	2.02	2.08	1.03	88	65
uSP	CADM1	1.00	-0.20	-0.20	NA	NA
uSP	NUTF2	0.81	-0.26	-0.32	>300	>300
uSP	GLG1	1.45	-0.53	-0.37	>300	>300
uSP	SERPINA1	1.47	-0.67	-0.46	>300	>300
uSP	CTSB	0.97	-0.51	-0.53	273	188
uSP	ITFG1	1.25	-0.74	-0.60	220	154
uSP	CST3	0.97	-0.70	-0.72	159	113
uSP	SPOCK1	2.50	-1.94	-0.77	141	101
uSP	MIEN1	0.93	-0.91	-0.98	95	69
uSP	PEBP4	1.36	-1.35	-0.99	94	69
uSP	KLK3	2.23	-3.08	-1.38	55	42
uEVP20	HBG2	1.92	2.66	1.39	58	44
uEVP20	OCIAD2	1.72	2.01	1.17	75	57
uEVP20	ITGAM	2.97	1.90	0.64	208	148
uEVP20	CARHSP1	1.13	0.36	0.32	>300	>300
uEVP20	RPLP1	1.35	0.19	0.14	>300	>300
uEVP20	KRT78	3.84	0.44	0.11	>300	>300

uEVP20	MB	1.65	-0.64	-0.39	>300	>300
uEVP20	RPS20	1.09	-1.06	-0.97	104	77
uEVP150	CKAP5	3.08	0.50	0.16	NA	NA
uEVP150	TACC2	3.60	0.18	0.05	NA	NA
uEVP150	GSTM4	4.09	-0.29	-0.07	NA	NA

Rebuttal Table 2: Estimated cohort sizes needed for robust determination of observed effect sizes in cISUP Grade Group signatures. Number of genes for each fraction: 3238 (uSP), 5686 (uEV-P20) and 4067 (uEV-P150). Assumed population prevalence of 0.57 in the largest group. NA: Standardized fold change is too small (<0.2) to estimate a sample size. Standardized fold change is calculated as the difference between groups divided by the within-class standard deviation, on the log2 scale. Tolerance: "The sample size n is chosen to ensure that the expected (average) accuracy of the classifier is within the tolerance of the best possible classifier". Source: National Cancer Institute Biometric Research Program guidelines for developing classifiers <https://brb.nci.nih.gov/brb/samplesize/samplesize4GE.html>).

We added the following text to the **Results** on **page 12-13 lines 246-255 and 260-263**:

"To create biomarkers of prostatic disease, we focused on proteins that were frequently detected (in >50% of samples), enriched in a urinary fraction, prostate-derived, and longitudinally stable (ICC > 0.4) (**Figure 6a**) This filtering strategy retained 226 uSP proteins, 280 uEV-P20 proteins and 235 uEV-P150 proteins. We used statistical machine learning to create and validate classifiers independently for each urine fraction. First, we created classifiers that distinguish cancer from non-cancer based solely on urine proteins; these had AUCs ranging from 0.92-1, significantly outperforming serum PSA (**Figure 6b**). In an independent, prospective validation cohort of 30 patients with proteomic data from all three urinary fractions (**Supplementary Table 1**), proteins had concordant effect sizes and validation AUCs ranging from 0.71-0.81 (**Figures 6c-d**)"

"In two independent validation cohorts of 199 and 75 patients (**Supplementary Table 1**), effect sizes of uSP proteins between the discovery and validation cohorts were concordant (**Supplementary Fig. 6b**)."

We added the following text to the **Discussion** on **page 17-18 lines 356-366**:

"Since our unique dataset represents one of the largest urine proteomics studies for prostate cancer, we were able to leverage urinary fraction-specificity, frequency of detection in prostate tissues and longitudinal stability to prioritize proteins as new biomarker candidates for prostatic diseases. The proteomics dataset generated in this study comprises one of the larger-sized discovery cohorts for urinary proteomics studies totaling >100 patients for prostate cancer biomarker discovery in urine, even for urinary EVs, and is well powered to detect log₂ fold changes in protein abundances of 1. However, while many of the proteomic differences between groups (e.g. cISUP Grade Groups) are concordant in independent cohorts, these differences are small. Further validation of the urinary signatures using robust quantitation by targeted proteomics assays with stable isotope labeled standards^{2,50} in larger, racially diverse validation cohorts is needed."

For verifying the uEV markers identified in **Figure 7c-d** that were differentially expressed in PCa vs. NC patients and had predicted cell surface localization, we assessed their fold changes in an independent cohort of 30 patients (Validation 1) described above in **Figure 6**. We show that for

all of the potential PCa/NC uEV surface markers, their differential expression in PCa vs. NC patients is conserved in Discovery and Validation 1 cohorts (**Figure 7d**).

Figure 7d: Frequency of detection (left panel) and differential abundance (\log_2FC , middle panel) of uEV surface markers in patients with prostate cancers (PCa) or no prostate cancers (NC) in the discovery cohort (151-153 patients) and an independent prospective validation cohort (Validation 1: 30 patients).

Reviewer #3 (Remarks to the Author):

The study provides comprehensive proteome profiles from multiple different urine fractions including soluble, small, and large urinary EVs from patients and EVs from prostate cancer cells. They assessed potential biases in EV biophysical characteristics and tissue specific protein expressions by DRE using pre- and post- urine samples. This showed no significant biases and increases of prostate specific proteins as physically perturbed prostate. uEV from patient urines display better correlation with prostate tissue context compared to soluble fractions and cEV proteome. In addition, proteome from the large uEV (uEV-P20) exhibits better correlation with prostate and disease specific contexts due to a larger protein cargo. Selected marker candidates were also detected with independent affinity-based EV enrichment method to assess a potential of the markers for translation utility.

The study highlights that patient-derived EV should be prioritized in translation studies. However, soluble fractions should have possible indications including blood driven immune responses, secreted proteins from other organs. These could be an alternative indicator of the cancer related immune responses and disease complications.

We thank the reviewer for acknowledging the value of our investigation into urinary EVs. We agree with the reviewer that the urinary soluble fraction is also interesting for biomarker discovery and therefore placed equal emphasis on the urinary soluble proteins in addition to the urinary EV proteins.

The study used proteome information perturbed by DRE due to the simplicity to enrich prostate proteins. However, this is not application in RP condition. Thus, it is more interesting to see whether the selected marker candidates exhibit similar level of significance without DRE.

In our study we were most interested in being able to risk-stratify patients pre-treatment. As such, all of the patients in this study are treatment-naïve at the time of urine collection and receive radical prostatectomy (RP) or radiotherapy later.

The study confirms utility of urinary EVs in translational and biomarker studies through a comprehensive proteome analysis by analyzing multiple different fractions. This actually increased the comprehension of the proteome from different fractions and exhibits proteome-wide heterogeneity in each fraction.

We thank the reviewer for acknowledging the value of our work.

The proteome from patients were not compared with benign prostatic conditions such as BPH. For example, Elevated levels of certain urinary proteins like PCA3 are not specific to prostate cancer and can also be seen in other benign prostate conditions. The prostate specific proteins are likely detected in benign conditions, as well.

We apologize for the confusion as we did include patients with BPH in the non-cancer cohort for all comparisons of prostate cancer patients to non-cancer patients.

We have clarified the text in the **Results** on page 11 lines 223-226 to emphasize that the non-cancer group contains patients with BPH. The text now reads:

“Men with benign prostatic conditions (non-cancer [NC]) included individuals with elevated serum PSA levels and benign prostatic hyperplasia as well as patients with no diagnosed prostate cancer on transrectal ultrasound-guided 12-core biopsy (**Supplementary Table 1**).”

We have also clarified the text in the **Experimental model and subject details** on page 39 to emphasize that the non-cancer group contains patients with BPH. The text now reads:

“Men with benign prostatic conditions (non-cancer [NC]) (**Supplementary Table 1**) included individuals with elevated serum PSA (sPSA) levels and benign prostatic hyperplasia (BPH; 44 patients; median sPSA 6.3 ng/mL, range 1.7 – 11.9 ng/mL) or no diagnosed prostate cancer on transrectal ultrasound-guided 12-core biopsy (Biopsy-negative; 20 patients; median sPSA 5.2 ng/mL, range 0.5 – 31.5 ng/mL). Selection criteria for men with benign prostatic conditions included a diagnostic sPSA level < 20 ng/mL and post-surgery sPSA level <0.1 ng/mL to exclude highly metastatic men. Other clinical details are detailed in **Supplementary Table 1**.”

And lack of validation with independent patient samples.

We agree with the reviewer that additional validation of these classifier models in independent patient cohorts is indeed necessary. We have acquired three independent validation cohorts for the validation of the PCa vs NC models for all three urine fractions (Validation cohort 1: 30 patients) and validation of the GG1 vs GG>1 uSP classifier (Validation cohort 2: 199 patients and Validation cohort 3: 70 patients).

In this revision, Reviewer 2 suggested that we did not fully leverage our proteomic data that was collected on discrete urinary fractions to extract information that could better inform biomarker discovery. We agree with the reviewer that we could have used a more stringent selection process

and have made changes to our biomarker discovery pipeline to include this information. Due to the new figures generated, the original **Figure 5** has been split into new **Figures 5 and 6**.

Our modified biomarker discovery pipeline now includes several new inclusion criteria (**Figure 6a**) for selecting the most relevant proteins for machine learning. In addition to the existing inclusion criteria of protein detection in >50% of urines and longitudinally stable proteins, our new inclusion criteria include enrichment of a protein in a specific urinary fraction (from differential expression analysis in **Figure 2j**), as well as protein detection in more than half of prostate tissues (suggesting that the protein is likely prostate-derived). These resulting proteins (uSP: 228, uEV-P20: 281, uEV-P150: 236) were then used as features for building classifiers by machine learning (new **Figure 6a**).

Figure 6a: New pipeline for biomarker discovery.

For validating prostate cancer/non-cancer signatures, we acquired urines from a cohort of 30 patients, of which 16 had prostate cancer and 14 had benign prostatic hyperplasia (non-cancer). We report validation AUCs from 0.71-0.81 and concordant effect sizes (\log_2 fold change) between discovery and validation datasets for all urinary fractions (**Figures 6b-d**).

Figures 6b-d: Performance of PCa/NC classifiers in discovery (B) and validation (D) datasets with effect sizes (C) of proteins in the classifiers.

We recognize that the size of validation cohort is small and we would like to note that BPH post-DRE urines are not typically bio-banked in prostate cancer clinics. In addition, for uEV proteomics analysis, we require 14 mL of urine per patient for uEV isolation, which is the entirety of the average volume of first-catch first-flow post-DRE urine (~15-20 mL). These large volumes of post-DRE urine are rarely available and only accessible after years of urine collection.

For verifying cISUP GG>1 vs GG=1 classifiers, we assembled independent validation cohorts consisting of 199 patients (Validation 2) and 70 patients (Validation 3) with prostate cancer (**Supplementary Table 1**). These samples were collected, processed and data acquired at two different institutes (Toronto or Norfolk, Virginia). We report concordant effect sizes for 5 proteins in 2/3 datasets and 3 proteins in 3/3 datasets (**Supplementary Fig. 6b**). The remaining proteins had small effect sizes in the discovery cohort. Our power analysis revealed that we were underpowered to validate the remainder of the proteins due to the limited size of the cohort (**Rebuttal Table 2**). Further validation of these candidate proteins in larger cohorts >300 samples will be required, should these larger cohorts be accessible in the future. Due to the large volumes of urine and cohort size required for validation of the GG1 vs GG>1 classifier for uEVs we were not able to validate that classifier at this point in time. We further discuss this point in our **Discussion** page 17-18 where we highlight the need for validation in larger, diverse cohorts and the need for validation using targeted mass spectrometry assays.

Supplementary Fig. 6b: Effect sizes of proteins in uSP cISUP>1/cISUP=1 protein signature in discovery and validation cohorts.

Classifier	gene	sd (log2)	fc (log2)	Standardized fc (log2)	Sample size needed (Tolerance = 0.05)	Sample size needed (Tolerance = 0.10)
uSP	CAT	2.02	2.08	1.03	88	65
uSP	CADM1	1.00	-0.20	-0.20	NA	NA
uSP	NUTF2	0.81	-0.26	-0.32	>300	>300
uSP	GLG1	1.45	-0.53	-0.37	>300	>300
uSP	SERPINA1	1.47	-0.67	-0.46	>300	>300
uSP	CTSB	0.97	-0.51	-0.53	273	188
uSP	ITFG1	1.25	-0.74	-0.60	220	154
uSP	CST3	0.97	-0.70	-0.72	159	113
uSP	SPOCK1	2.50	-1.94	-0.77	141	101
uSP	MIEN1	0.93	-0.91	-0.98	95	69
uSP	PEBP4	1.36	-1.35	-0.99	94	69
uSP	KLK3	2.23	-3.08	-1.38	55	42
uEVP20	HBG2	1.92	2.66	1.39	58	44
uEVP20	OCIAD2	1.72	2.01	1.17	75	57
uEVP20	ITGAM	2.97	1.90	0.64	208	148
uEVP20	CARHSP1	1.13	0.36	0.32	>300	>300
uEVP20	RPLP1	1.35	0.19	0.14	>300	>300
uEVP20	KRT78	3.84	0.44	0.11	>300	>300
uEVP20	MB	1.65	-0.64	-0.39	>300	>300
uEVP20	RPS20	1.09	-1.06	-0.97	104	77

uEVP150	CKAP5	3.08	0.50	0.16	NA	NA
uEVP150	TACC2	3.60	0.18	0.05	NA	NA
uEVP150	GSTM4	4.09	-0.29	-0.07	NA	NA

Rebuttal Table 2: Estimated cohort sizes needed for robust determination of observed effect sizes in cISUP Grade Group signatures. Number of genes for each fraction: 3238 (uSP), 5686 (uEV-P20) and 4067 (uEV-P150). Assumed population prevalence of 0.57 in the largest group. NA: Standardized fold change is too small (<0.2) to estimate a sample size. Standardized fold change is calculated as the difference between groups divided by the within-class standard deviation, on the log₂ scale. Tolerance: "The sample size n is chosen to ensure that the expected (average) accuracy of the classifier is within the tolerance of the best possible classifier" (from NCI Biometric Research Program guidelines for developing classifiers).

We added the following text to the **Results** on **page 12-13 lines 246-255 and 260-263**:

"To create biomarkers of prostatic disease, we focused on proteins that were frequently detected (in >50% of samples), enriched in a urinary fraction, prostate-derived, and longitudinally stable (ICC > 0.4) (**Figure 6a**) This filtering strategy retained 226 uSP proteins, 280 uEV-P20 proteins and 235 uEV-P150 proteins. We used statistical machine learning to create and validate classifiers independently for each urine fraction. First, we created classifiers that distinguish cancer from non-cancer based solely on urine proteins; these had AUCs ranging from 0.92-1, significantly outperforming serum PSA (**Figure 6b**). In an independent, prospective validation cohort of 30 patients with proteomic data from all three urinary fractions (**Supplementary Table 1**), proteins had concordant effect sizes and validation AUCs ranging from 0.71-0.81 (**Figures 6c-d**)"

"In two independent validation cohorts of 199 and 75 patients (**Supplementary Table 1**), effect sizes of uSP proteins between the discovery and validation cohorts were concordant (**Supplementary Fig. 6b**)."

We added the following text to the **Discussion** on **page 17-18, lines 356-366**:

"Since our unique dataset represents one of the largest urine proteomics studies for prostate cancer, we were able to leverage urinary fraction-specificity, frequency of detection in prostate tissues and longitudinal stability to prioritize proteins as new biomarker candidates for prostatic diseases. The proteomics dataset generated in this study comprises one of the larger-sized discovery cohorts for urinary proteomics studies totaling >100 patients for prostate cancer biomarker discovery in urine, even for urinary EVs, and is well powered to detect log₂ fold changes in protein abundances of 1. However, while many of the proteomic differences between groups (e.g. cISUP Grade Groups) are concordant in independent cohorts, these differences are small. Further validation of the urinary signatures using robust quantitation by targeted proteomics assays with stable isotope labeled standards^{2,50} in larger, racially diverse validation cohorts is needed."

There is discrepancy in EV selectivity and protein detection sensitivity between centrifugation-based and affinity-based methods.

We agree with the reviewer that EV selectivity, protein detection and protein abundance can differ depending on the choice of EV isolation method. We have added the following text to the **Discussion** on **page 17-18, lines 349-355**, to point out this caveat.

“In this study we used differential ultracentrifugation, which is a less specific EV isolation approach compared to affinity-based capture or differential gradient ultracentrifugation which can isolate EV populations with higher specificity⁵¹. As such, the selectivity, protein composition and protein abundance in EVs may differ depending on the choice of EV isolation method. Future experiments will be needed to verify if these putative markers are able to capture similar EV subpopulations.

Factors like hydration status, time of day, and differences in collection and storage can affect urinary protein levels and make results less consistent. These points should be at least discussed or addressed by the experiments at some part.

We agree with the reviewer that all these factors can affect urinary protein levels. We did not perform experiments to assess factors like hydration status, time of day, and differences in collection and storage as these factors have been thoroughly discussed by the field. We refer to these studies as well as our assessment of longitudinal stability of urinary proteins (**Figure 5a-b**) in the **Discussion** on **page 16** which reads:

“Prior work has rigorously quantified the role of factors such as sample collection, processing^{2,43,44} and storage⁴⁵ on urine proteomes. We show that digital rectal examinations are a simple way to enrich for prostate proteins and EVs in first-catch urine. While post-DRE urine proteomes can vary over time^{4,46}, a subset of specific proteins are temporally stable over many months and are well suited for non-invasive sampling.”

To isolate urinary EVs in this study, we used a published protocol by Correll et al⁵ who assessed the reproducibility of EV isolation from post-DRE urine and found it to be highly reproducible Spearman’s $\rho > 0.99$. We also independently repeated their protocol using independent post-DRE urine samples and saw similar results. We assessed reproducibility in five post-DRE urine samples - 3 samples from 3 patients with cISUP Grade Group 1 prostate cancer, 1 pooled sample from 10 non-cancer patients with benign prostatic hyperplasia, and 1 pooled sample from 10 non-cancer patients who had elevated serum PSA levels (serum PSA > 4 ng/mL) but no cancer detected on transrectal ultrasound-guided needle biopsy). In our cohort, the median Spearman’s ρ for processing replicates for uEV-P20 and uEV-P150 was 0.941 (range 0.890 – 0.975) and 0.948 (range 0.925 – 0.972), respectively. Proteomes of processing replicates were more similar to each other than between biological groups (e.g. Prostate cancer *versus* non-cancer), suggesting that our uEV isolation protocol is highly reproducible and able to uncover differences between clinical groups.

We have added this analysis to **Supplementary Fig. 3**. We have also updated the main text on **page 8, lines 143-146** to read:

“uEV isolation was also highly reproducible, with larger proteomic differences observed between patients and clinical groups compared to within samples (**Supplementary Fig. 3a-b**).”

A**B**
Supplementary Fig. 3: Reproducibility of uEV isolation from post-DRE urine.

REVIEWERS' COMMENTS

Reviewer #1 (Remarks to the Author):

In their revision, the authors have responded adequately to most of my comments. They provided additional validation of their proposed "prostate cancer vs. non-cancer" urine biomarkers in an independent validation cohort. They also assembled two additional independent validation cohorts for verifying the "GG>1 vs. GG=1" classifier, and they validated the effect sizes of some but not all of the uSP proteins, noting that larger sample sizes would be needed that were not available for full validation. They also provided further analyses, including that of urine EV proteins unique to this study with differential abundance in cancer vs. non-cancer, and additional data on reproducibility of their assay. They have also modified the text to further discuss many of the important limitations of the study.

Reviewer #3 (Remarks to the Author):

Authors addressed all the concerns through this revised manuscript. No other concerns were made from my side.

RESPONSE TO REVIEWERS' COMMENTS

Reviewer #1 (Remarks to the Author):

In their revision, the authors have responded adequately to most of my comments. They provided additional validation of their proposed "prostate cancer vs. non-cancer" urine biomarkers in an independent validation cohort. They also assembled two additional independent validation cohorts for verifying the "GG>1 vs. GG=1" classifier, and they validated the effect sizes of some but not all of the uSP proteins, noting that larger sample sizes would be needed that were not available for full validation. They also provided further analyses, including that of urine EV proteins unique to this study with differential abundance in cancer vs. non-cancer, and additional data on reproducibility of their assay. They have also modified the text to further discuss many of the important limitations of the study.

We thank the reviewer for this positive evaluation of our revised manuscript.

Reviewer #3 (Remarks to the Author):

Authors addressed all the concerns through this revised manuscript. No other concerns were made from my side.

We thank the reviewer for this positive evaluation of our revised manuscript.